# MILCO: Learned Sparse Retrieval Across Languages via a Multilingual Connector

**Thong Nguyen, Yibin Lei & Jia-Huei Ju**
University of Amsterdam
{t.nguyen2,y.lei,j.ju}@uva.nl

**Eugene Yang & Andrew Yates**
Johns Hopkins University, HLTCOE
{eugene.yang, andrew.yates}@jhu.edu

## Abstract

Learned Sparse Retrieval (LSR) combines the efficiency of bi-encoders with the transparency of lexical matching, but existing approaches struggle to scale beyond English. We introduce MILCO, an LSR architecture that maps queries and documents from different languages into a shared English lexical space via a multilingual connector. MILCO is trained with a specialized two-stage regime that combines Sparse Alignment Pretraining with contrastive training to provide representation transparency and effectiveness while mitigating semantic collapse. Motivated by the observation that uncommon entities are often lost when projected into English, we propose a new LexEcho head, which enhances robustness by augmenting the English lexical representation with a source-language view obtained through a special [**ECHO**] token. MILCO achieves state-of-the-art multilingual and cross-lingual LSR performance, outperforming leading dense, sparse, and multi-vector baselines such as BGE-M3 and Qwen3-Embed on standard multilingual benchmarks, while supporting dynamic efficiency through post-hoc pruning. Notably, when using mass-based pruning to reduce document representations to only 30 active dimensions on average, MILCO 560M outperforms the similarly-sized Qwen3-Embed 0.6B with 1024 dimensions, while achieving $3\times$ lower retrieval latency and $10\times$ smaller index size.[1]

## 1 Introduction

Learned Sparse Retrieval (LSR) represents queries and documents as sparse lexical embeddings and retains the scalability benefits of bi-encoders (MacAvaney et al., 2020; Formal et al., 2021; Nguyen et al., 2023) . Unlike dense methods, LSR aligns representation with a natural language vocabulary, yielding transparent representations that facilitate error tracing and bias inspection. LSR naturally supports dynamic post-hoc pruning at inference time (Bruch et al., 2024), providing Matryoshka-like latency control (Kusupati et al., 2022) without requiring auxiliary training objectives. Empirically, LSR (Lassance et al., 2024; Lei et al., 2025) is competitive on benchmarks like BEIR (Thakur et al., 2021) and MTEB (Enevoldsen et al., 2025). Theoretically, recent work shows sparse lexical embeddings exhibit higher representational capacity than dense embeddings, which is illustrated by their superior performance on the LIMIT benchmark (Weller et al., 2025) where even state-of-the-art dense models fail catastrophically.

Thus far, LSR progress has been driven primarily by English (Formal et al., 2022; Shen et al., 2025; Nardini et al., 2025), where models such as SPLADE (Lassance et al., 2024) deliver strong zero-shot effectiveness and have seen wide adoption in production systems (e.g., OpenSearch, ElasticSearch, Sentence Transformers). Extensions beyond English remain fragmented: BGE-M3 (Chen et al., 2024) combines dense, sparse, and multi-vector heads under a shared backbone, but its sparse component underperforms and lacks cross-lingual support; conversely, SPLADE-X (Nair et al., 2022b)

---

[1]Our code is available at: https://github.com/thongnt99/milco.

and BLADE (Nair et al., 2023) target cross-lingual retrieval only and rely on training separate models for each language pair, limiting their applications.

A straightforward multilingual LSR approach is to attach a multilingual MLM head to a multilingual base encoder, projecting inputs into the full multilingual vocabulary. However, directly optimizing such models can lead to severe semantic collapse (Nguyen et al., 2024), where representations lose interpretable term semantics, resulting in significant degradation of the model's transparency and effectiveness. This behavior is demonstrated both qualitatively and quantitatively in Section 5.

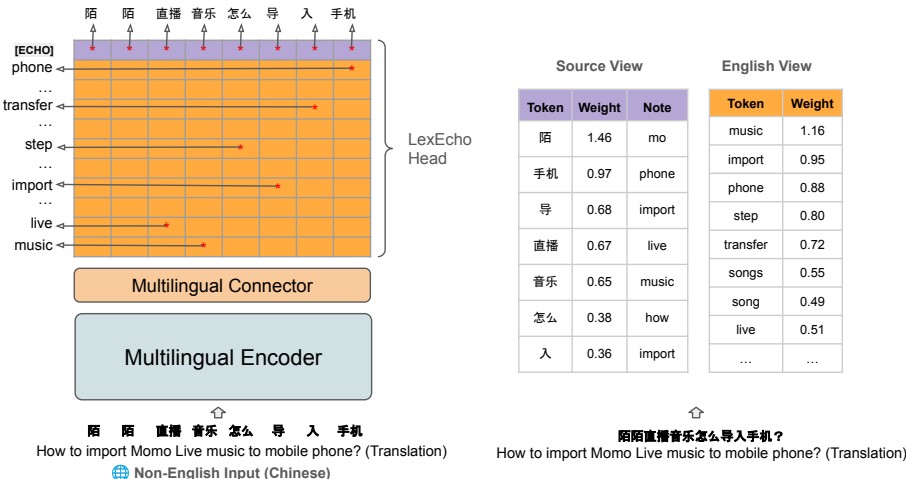

Figure 1: MILCO's LexEcho head produces two lexical views: (1) a pivot (English) view supporting cross-lingual and multilingual retrieval, and (2) a source view for robustness to uncommon entities.

To overcome those challenges, we introduce MILCO, illustrated in Figure 1, an LSR architecture that uses a multilingual connector between a multilingual base encoder and an English MLM head, mapping text from all languages into a shared English vocabulary space. MILCO collapses the multilingual vocabulary to English to create a universal representation, which also reduces memory and computation during training. This approach enables one single MILCO model to support both multilingual and cross-lingual retrieval across many languages.

**MILCO Training.** We adopt a two-stage training procedure. First, we propose *Sparse Alignment Pretraining (SAP)*, which maps multilingual inputs to English lexical targets, in contrast to prior dense alignment methods that operate in low-dimensional latent space (Reimers & Gurevych, 2020). SAP leverages widely available bitext corpora instead of scarce multilingual relevance labels, enabling large-scale multilingual pretraining. Alignment pretraining enables the model to then be fine-tuned with contrastive training using distillation (Lassance et al., 2024), which enhances retrieval effectiveness while preserving grounding. Crucially, SAP is a prerequisite: without alignment, contrastive training leads to semantic collapse, harming effectiveness.

**LexEcho Head.** We observe that uncommon entities, especially from non-Latin languages, are often lost when projected into English. To address this, we introduce LexEcho, a dual-view LSR head illustrated in Figure 1. The *pivot (English) view* is obtained by max-pooling over the logit matrix of an English MLM head, with our multilingual connector enabling it to operate across many languages. The *source view* selectively echoes input tokens through a special [**ECHO**] token, preserving entities that the English view fails to capture and assigning higher scores to more important tokens. This approach allows the model to represent entities it has never seen before or cannot translate.

Across 39 languages, our 560M MILCO model sets a new state of the art for Learned Sparse Retrieval in both multilingual and cross-lingual settings. On MIRACL, our best model surpasses BGE-Sparse, BGE-Dense, and Qwen3-Embed 8B by +34.1%, +4.5%, and +3.6% nDCG@10, respectively, while also providing transparent representations. Experiments also show that the proposed LexEcho head enhances robustness to tail entities, yielding an +4.2% overall improvement on MIRACL. Like Matryoshka Representation Learning, MILCO supports controllable efficiency via post-

hoc pruning, surpassing Qwen3-Embed 0.6B with only 30 active dimensions per document, while achieving $3\times$ lower retrieval latency and $10\times$ smaller index size.

**Our Contributions:**

- We introduce MILCO, a multilingual connector architecture that maps queries and documents into a shared English lexical space, unifying multilingual and cross-lingual retrieval within a single model. Its LEXECHO head provides dual lexical views, enhancing robustness to unseen or uncommon entities or concepts.
- We introduce a new Sparse Alignment Pretraining (SAP) pretraining strategy tailored to multilingual LSR that addresses semantic collapse and provides the foundation for contrastive training, leading to an effective and transparent model.
- Through comprehensive experiments on multilingual and cross-lingual benchmarks across 39 languages, we demonstrate that the MILCO architecture and Sparse Alignment Pretraining are key to achieving state-of-the-art multilingual and cross-lingual sparse retrieval.

## 2 RELATED WORK

**Learned Sparse Retrieval (LSR).** Zamani et al. (2018) first proposed SNRM, an n-gram neural model for learning sparse representations compatible with inverted indexes, though its representations remained latent. Subsequent work (MacAvaney et al., 2020; Formal et al., 2021) replaced SNRM with Transformer architectures that map text directly into the English lexicon, yielding more transparent and effective models. Nguyen et al. (2023) categorize LSR architectures into three groups: Binary Encoders, which assign binary weights to tokens and enable efficient inference-free query encoding with modest effectiveness trade-offs (Nardini et al., 2025; Shen et al., 2025); MLP Encoders, which score tokens by contextual importance (MacAvaney et al., 2020; Lin & Ma, 2021); and MLM Encoders, used in state-of-the-art methods like Splade (Formal et al., 2021), which provide differentiable query weighting and expansion. Beyond architecture, training protocols such as hard negative mining and distillation (e.g., from cross-encoders) are key to narrowing the gap with dense and hybrid systems (Formal et al., 2022; Lassance et al., 2024). In this work, we introduce MILCO, a new LSR architecture with a LexEcho head for multilingual sparse retrieval.

**Multilingual/Cross-language Retrieval.** A central challenge in cross-language IR is the language mismatch between queries and documents. Existing approaches address this either through translation pipelines or multilingual encoders that map text from different languages into a shared latent space for cross-lingual matching. Representative efforts include dense encoder methods (Zhang et al., 2024; Wang et al., 2024; Zhang et al., 2025) and multi-vector methods with multilingual pretraining (Louis et al., 2024; Yang et al., 2024a). Community benchmarks such as MIRACL (Zhang et al., b) and NeuCLIR (Lawrie et al., 2024) provide standardized evaluation across many languages, while studies on translationese highlight biases introduced by translated text (Gellerstam, 1986; Riley et al., 2020; Nair et al., 2022a; Zhang et al., a). For sparse retrieval, BGE-M3 (Chen et al., 2024) combines dense, sparse, and multi-vector heads for multilingual retrieval, but its sparse component underperforms and offers limited cross-language support. Other sparse models such as SPLADE-X (Nair et al., 2022b) and BLADE (Nair et al., 2023) focus on cross-language retrieval with language-specific models. In contrast, our sparse model, MILCO, supports both multilingual and cross-language retrieval within a single model while substantially outperforming prior approaches.

**Alignment Pretraining.** Previous work highlights the importance of multilingual pre-training for building shared cross-language semantic spaces (Conneau et al., 2020; Chi et al.; Feng et al., 2022; Yang et al., 2022). For retrieval, pre-training directly on relevance objectives has been explored, often using in-batch negatives and hard-negative mining (Zhang et al., 2024). Another direction focuses on distilling efficient models, where cross-encoder or ensemble teachers guide bi-encoder students to produce retrieval-friendly embeddings (Kim et al., 2023; Campos et al., 2023). In multilingual IR, distillation also yields compact, language-agnostic dense embeddings for scalable cross-language retrieval (Reimers & Gurevych, 2020; Yang et al., 2024a). While prior work has mainly focused on dense models, we are the first to explore multilingual sparse alignment and introduce a sparse alignment pre-training method that enables LSR to perform well on multilingual data.

# 3 PROPOSED METHODOLOGY

## 3.1 MILCO ARCHITECTURE

MILCO consists of three main components: (i) a **Multilingual Encoder**, (ii) a **Multilingual Connector**, and (iii) a **LexEcho Head**. Figure 1 illustrates MILCO processing the Chinese input: *"How to import Momo Live music to a mobile phone?"* Let $\mathcal{L}$ denote the set of supported languages. For an input text $x$ in language $\ell \in \mathcal{L}$, we first tokenize it into a sequence of $n$ source tokens:

$$\mathbf{s}^{(\ell)} = (s_1, \ldots, s_n) \tag{1}$$

**Multilingual Encoder.** A transformer-based Multilingual Encoder $Enc(\cdot)$ maps the input tokens $s^{(\ell)}$ into a sequence of hidden states of dimension $d_{\mathcal{L}}$ in a multilingual embedding space:

$$\mathbf{H}^{(\ell)} = Enc(\mathbf{s}^{(\ell)}) \in \mathbb{R}^{n \times d_{\mathcal{L}}} \tag{2}$$

where $\mathbf{H}^{(\ell)}$ represents the contextualized embeddings for the $n$ input tokens. For conciseness, we omit the superscript $(\ell)$ whenever the language space of the variable is unambiguous, making it $\mathbf{H}$.

**Multilingual Connector.** The Multilingual Connector $\phi$ then projects these multilingual hidden states $\mathbf{H}$ into $\mathbf{Z}$ of dimension $d_e$, which live in the embedding space of the pivot language:

$$\mathbf{Z} = \text{LayerNorm}\left(\text{Linear}(\phi(\mathbf{H})\right) \in \mathbb{R}^{n \times d_e}, \quad \text{where } \phi(\cdot) : \mathbb{R}^{n \times d_{\mathcal{L}}} \rightarrow \mathbb{R}^{n \times d_e} \tag{3}$$

For simplicity, we implement the connector $\phi$ with a Multi-Layer Perceptron. This projection unifies representations across different languages through English as the pivot, allowing our LexEcho Head to project them into a shared English lexicon. While architecturally there is no restriction on the selection of the pivot language, we select English because of its rich resources and the availability of LSR teacher models for alignment, which we discuss later in this section.

**LexEcho Head.** The LexEcho head produces a dual-view lexical representation from the projected states $\mathbf{Z}$. It generates two complementary sparse views: ① an *Pivot (English) View* that captures semantic concepts in English and ② a *Source View* that preserves important source input tokens.

① *Pivot (English) View*: The English lexical representation is generated by an English MLM head, as in LSR models like SPLADE (Lassance et al., 2024), but our multilingual connector extends this to the 39+ languages supported by our base model.

Multilingual representations $\mathbf{Z}$ (Eq. 3) are linearly refined and decoded onto the English vocabulary $V_e$ via an embedding matrix $\boldsymbol{E} \in \mathbb{R}^{|V_e| \times d_e}$ and bias $\mathbf{b}_v$, yielding logits that score each source token against every English token.

$$\mathbf{T}^{(e)} = \log\left(1 + \text{ReLU}\left(Dec(\mathbf{Z})\right)\right) \in \mathbb{R}_{\geq 0}^{n \times |V_e|}, \quad \text{where } Dec(\mathbf{x}) = \mathbf{x}\boldsymbol{E}^{\top} + \mathbf{b}_v. \tag{4}$$

Here, we define the log-saturation effect function, introduced by MacAvaney et al. (2020); Formal et al. (2021) as $\text{LogSat}(\cdot)$ for simplicity,

$$\text{LogSat}(\mathbf{x}) = \log(1 + \text{ReLU}(\mathbf{x})) \tag{5}$$

Next, max-pooling across source tokens ($n$) yields the final English lexical representation:

$$\mathbf{t}^{(e)} = \left(\max_i \mathbf{T}_{i1}^{(e)}, \max_i \mathbf{T}_{i2}^{(e)}, ..., \max_i \mathbf{T}_{i|V_e|}^{(e)}\right), \quad \text{where } i \in [1, n] \tag{6}$$

This English view $\mathbf{t}^{(e)}$ is sparse and includes not only direct translations (*live*, *music*, *phone*) but semantically related terms (*song*, *stream*, *step*) that supports semantic retrieval.

② *Source View*: The connector maps common concepts into English but can fail on uncommon or unseen entities, especially in non-Latin scripts (e.g., *Momo* in Figure 1), or when names differ across languages (e.g., *Douyin* vs. *TikTok*). Scaling model size alone cannot solve this, as new entities continually appear.

Our LexEcho head tackles this by selectively echoing key tokens from the source. A dedicated [**ECHO**] token in the MLM head, denoted $Dec_{[\textbf{ECHO}]}(\cdot)$, produces a weight vector $\mathbf{w}$ for each source token to ensure crucial tokens are selected:

$$\mathbf{w} = \text{LogSat}\left(Dec_{[\textbf{ECHO}]}\left(\mathbf{Z}\right)\right) \in \mathbb{R}^n_{\geq 0} \tag{7}$$

By combining the English $\mathbf{t}^{(e)}$ and the weighted source views $\{\mathbf{s}_i^{(l)}, \mathbf{w}_i\}_{i=1}^n$, MILCO produces a dual-view representation $\mathbf{o} = \{\mathbf{t}^{(e)}, \mathbf{s}^{(l)}, \mathbf{w}\}$ that leverages cross-lingual projection to form a unified lexical view preserving crucial source-language tokens that would otherwise be lost in translation.

### 3.2 Training: Sparse Alignment and Contrastive Refinement

We propose a two-stage training recipe for MILCO: Sparse Alignment Pretraining to ground multilingual text to English lexical space, followed by Sparse Contrastive Training to refine alignment and optimize retrieval effectiveness, with sparsity enforced throughout.

**Sparse Alignment Pretraining (SAP).** To ensure the English view $\mathbf{t}^{(e)}$ is grounded in the English lexicon, we leverage widely available parallel (xx–en) sentences to align the English view of a non-English sentence to the representation of its corresponding English sentence. Given a pair of tokenized parallel sentences $(\mathbf{s}^{(\ell)}, \mathbf{s}^{(e)})$ in language $\ell$ and English, we employ an oracle teacher English LSR model, such as SPLADEv3 (Lassance et al., 2024), denoted as $\text{LSR}^*$, to produce the target English sparse representation $\mathbf{t}^*$.

We design a sparse-aware MSE (SMSE) loss, specifically to minimize the difference between two sparse vectors. Since most coordinates are zero, the learning signal should concentrate on the few active ones. Also, with the $\text{LogSat}(\cdot)$ activation, negative pre-activation values yield zero gradients. Therefore, we compute the loss directly on the decoded logits, i.e. $Dec(\mathbf{Z})$, which were the input to $\text{LogSat}(\cdot)$, with max-pooling across the input tokens and restrict it to coordinates where at least one side is positive. For clarity, we denote such augmented representations as $\tilde{\mathbf{t}}^{(e)}$ and $\tilde{\mathbf{t}}^*$. Formally, the SMSE loss can be written as

$$L_{\text{SMSE}}\left(\mathbf{t}^{(e)}, \mathbf{t}^*\right) = \frac{\sum_{j=1}^{|V_e|} \mathbf{1}\left(\tilde{\mathbf{t}}_j^{(e)} > 0 \ \vee \ \tilde{\mathbf{t}}_j^* > 0\right)\left(\tilde{\mathbf{t}}_j^{(e)} - \tilde{\mathbf{t}}_j^*\right)^2}{\sum_{j=1}^{|V_e|} \mathbf{1}\left(\tilde{\mathbf{t}}_j^{(e)} > 0 \ \vee \ \tilde{\mathbf{t}}_j^* > 0\right)}, \tag{8}$$

where $\mathbf{1}(\cdot)$ denotes the indicator function. This SMSE objective mitigates gradient dilution and focuses training on informative lexical coordinates, yielding more stable alignment. During training, we apply SMSE over batches flattened into single vectors.

**Sparse Contrastive Training (SCT).** Alignment pretraining grounds multilingual inputs in a shared English lexicon but is not directly optimized for retrieval. To improve effectiveness, we further train MILCO with a LexEcho head using a contrastive objective on retrieval datasets. Following Lassance et al. (2024), we use a KL distillation loss (details in Section A.7) to transfer knowledge from a cross-encoder to MILCO. To promote sparsity, we add $\ell_1$-norm regularization on query and document representations $q$ and $p$. Concretely, the training objective is $L_{\text{contrastive}} = L_{\text{KLD}} + \alpha_q \|q\|_1 + \alpha_d \|p\|_1$, where the $\ell_1$-norms are implemented as means over the training batch.

## 4 Experimental settings

**Pretraining, Training and Evaluation Data.** For Sparse Alignment Pretraining, we use 594M bitext pairs from diverse domains collected with Sentence Transformers (Reimers & Gurevych, 2019), where each pair contains an English sentence and its translation. Dataset statistics are shown in Table 14. For Sparse Contrastive Training, we adopt the 1.4M multilingual queries released by Chen et al. (2024), with positive/negative documents and teacher scores obtained from bge-reranker-v2.5[2] reranker. More details are in Table 15.

---

[2]bge-reranker-v2.5-gemma2-lightweight

Following Chen et al. (2024), we evaluate MILCO on four benchmarks: MIRACL (Zhang et al., b), a large-scale multilingual retrieval benchmark covering 18 languages with high-quality human annotations; MTEB v2 (Enevoldsen et al., 2025) for large-scale multilingual retrieval; MLDR (Chen et al., 2024), a multilingual long-document retrieval benchmark in 13 languages; and MKQA (Long-pre et al., 2021), a cross-lingual benchmark with English documents and queries in 25 languages. Additional results on BEIR (Thakur et al., 2021), NeuCLIR (Lawrie et al., 2024) and LIMIT (Weller et al., 2025) are also included in the Appendix. Our evaluation spans 39 languages in total.

Table 1: Multilingual passage retrieval performance on the MIRACL dev set (measured by nDCG@10). Superscript $^*$: results obtained from Lassance (2023).

| Model | Size | Avg | ar | bn | en | es | fa | fi | fr | hi | id | ja | ko | ru | sw | te | th | zh | de | yo |
|---|---|---|---|---|---|---|---|---|---|---|---|---|---|---|---|---|---|---|---|---|
| *Dense, multi-vector and hybrid baselines* | | | | | | | | | | | | | | | | | | | | |
| mE5$_{large}$ | 560M | 66.6 | 76.0 | 75.9 | 52.9 | 52.9 | 59.0 | 77.8 | 54.5 | 62.0 | 52.9 | 70.6 | 66.5 | 67.4 | 74.9 | 84.6 | 80.2 | 56.0 | 56.4 | 78.3 |
| E5$_{mistral-7b}$ | 7.11B | 63.4 | 73.3 | 70.3 | 57.3 | 52.2 | 52.1 | 74.7 | 55.2 | 52.1 | 52.7 | 66.8 | 61.8 | 67.7 | 68.4 | 73.9 | 74.0 | 54.0 | 54.1 | 79.7 |
| M3-Dense | 560M | 69.2 | 78.4 | 80.0 | 56.9 | 56.1 | 60.9 | 78.6 | 58.3 | 59.5 | 56.1 | 72.8 | 69.9 | 70.1 | 78.7 | 86.2 | 82.6 | 62.7 | 56.7 | 81.8 |
| M3-Multi-vec | 560M | 70.5 | 79.6 | 81.0 | 59.3 | 57.8 | 62.0 | 80.1 | 59.4 | 61.5 | 58.3 | 74.5 | 71.2 | 71.2 | 79.1 | 87.9 | 83.0 | 63.7 | 58.0 | 82.4 |
| M3-Dense+Sparse | 560M | 70.4 | 79.6 | 80.7 | 58.8 | 58.1 | 62.3 | 79.7 | 58.0 | 62.9 | 58.3 | 73.9 | 71.2 | 69.8 | 78.5 | 87.2 | 83.1 | 63.5 | 57.7 | 83.3 |
| M3-Dense+Sparse+Multivector | 560M | 71.5 | 80.2 | 81.5 | 59.6 | 59.7 | 63.4 | 80.4 | 61.2 | 63.3 | 59.0 | 75.2 | 72.1 | 71.7 | 79.6 | 88.1 | 83.7 | 64.9 | 59.8 | 83.5 |
| PLAID-X (Multivector) | 560M | 55.5 | 66.0 | 68.0 | 46.4 | 51.4 | 48.3 | 52.5 | 61.9 | 42.8 | 56.8 | 44.6 | 61.2 | 63.4 | 61.2 | 32.9 | 75.6 | 72.0 | 44.5 | 49.1 |
| Qwen3-Embed - 0.6B | 596M | 60.5 | 69.9 | 66.3 | 51.5 | 54.2 | 52.7 | 69.7 | 54.4 | 51.3 | 51.4 | 63.3 | 60.1 | 59.7 | 48.6 | 77.2 | 73.8 | 58.3 | 52.9 | 74.0 |
| Qwen3-Embed - 8B | 7.57B | 69.8 | 78.2 | 78.3 | 59.8 | 59.6 | 60.5 | 79.0 | 61.0 | 63.1 | 56.1 | 74.3 | 67.5 | 73.5 | 72.2 | 84.3 | 81.5 | 63.3 | 60.5 | 84.5 |
| MILCO-dense (align + distill) | 560M | 67.9 | 77.0 | 76.6 | 55.3 | 57.5 | 60.2 | 77.1 | 59.0 | 60.5 | 55.5 | 70.5 | 70.9 | 67.5 | 74.4 | 86.1 | 80.6 | 62.5 | 57.6 | 72.8 |
| MILCO-dense (distill) | 560M | 70.9 | 79.5 | 80.2 | 56.8 | 60.4 | 63.4 | 78.5 | 62.6 | 62.2 | 58.4 | 74.7 | 70.5 | 72.1 | 79.6 | 87.0 | 82.9 | 64.2 | 59.5 | 83.2 |
| *Sparse baselines* | | | | | | | | | | | | | | | | | | | | |
| BM25 | 2 | 31.9 | 39.5 | 48.2 | 26.7 | 7.7 | 28.7 | 45.8 | 11.5 | 35.0 | 29.7 | 31.2 | 37.1 | 25.6 | 35.1 | 38.3 | 49.1 | 17.5 | 12.0 | 56.1 |
| T-Splade$^*$ | 3.4B | 54.5 | – | – | – | – | – | – | – | – | – | – | – | – | – | – | – | – | – | – |
| mSPLADEsTok$^*$ | - | 63.9 | – | – | – | – | – | – | – | – | – | – | – | – | – | – | – | – | – | – |
| OpenSearch[3] | 167M | | 74.0 | 67.0 | 57.5 | 54.2 | 51.4 | 76.7 | 55.8 | 48.6 | 58.2 | 66.9 | 60.7 | 65.8 | 76.8 | 74.0 | – | 56.2 | – | – |
| M3-Sparse | 560M | 53.9 | 67.1 | 68.9 | 43.8 | 38.6 | 45.1 | 65.4 | 35.3 | 48.2 | 48.9 | 56.1 | 61.5 | 44.5 | 57.9 | 79.1 | 70.9 | 36.1 | 32.5 | 70.0 |
| ① **MILCO** (SAP, SCT$_{KD}$, LexEcho) | 560M | **72.3** | **80.4** | **82.6** | **60.4** | **60.9** | 62.3 | **81.2** | 61.7 | **64.4** | 60.9 | **77.2** | 72.1 | 74.6 | 80.3 | 87.9 | 84.2 | 65.5 | **61.4** | **83.6** |
| ② **MILCO** (SAP, SCT$_{KD}$, MLM$_{en}$) | 560M | 69.4 | 77.3 | 79.5 | 57.6 | 59.7 | 58.5 | 78.8 | 60.6 | 63.4 | 57.7 | 72.8 | 67.7 | 72.6 | 78.1 | 82.3 | 80.4 | 60.6 | 59.7 | 81.2 |
| ③ **MILCO** (SAP, SCT, LexEcho) | 560M | 70.1 | 79.4 | 80.8 | 57.6 | 57.2 | 60.6 | 80.1 | 57.7 | 63.3 | 58.4 | 75.2 | 71.0 | 72.6 | 77.2 | 87.2 | 82.7 | 60.8 | 59.3 | 81.6 |
| ④ **MILCO** (SAP, MLM$_{en}$) | 560M | 54.5 | 59.8 | 59.7 | 57.0 | 56.0 | 44.9 | 66.0 | 48.2 | 58.6 | 48.8 | 54.9 | 59.4 | 51.2 | 47.8 | 55.0 | 55.9 | 46.7 | 48.5 | 62.2 |
| ⑤ **MILCO** (SCT$_{KD}$, MLM$_{en}$) | 560M | 59.2 | 72.7 | 72.3 | 47.7 | 47.6 | 50.9 | 72.5 | 48.8 | 50.4 | 51.8 | 64.0 | 62.7 | 53.6 | 62.3 | 77.7 | 72.8 | 46.2 | 44.8 | 66.2 |
| ⑥ **noMILCO** (SCT$_{KD}$, MLM$_{m}$) | 560M | 50.7 | 65.8 | 62.0 | 39.7 | 39.4 | 42.0 | 67.0 | 38.5 | 36.9 | 44.9 | 56.1 | 52.9 | 47.3 | 58.6 | 71.0 | 67.0 | 41.8 | 34.6 | 46.9 |

**Baselines.** We compare MILCO against two group of baselines: Dense/Multi-vector and Sparse methods. *For dense/multi-vector baselines*, we include recent state-of-the-art methods, including multilingual E5 (Wang et al., 2024), BGE-M3 (Chen et al., 2024), PLAID-X (Yang et al., 2024b), Qwen3 Embeddings (Zhang et al., 2025). *For sparse baselines*, we include unsupervised BM25, M3-Sparse (Chen et al., 2024) and also OpenSearch (Shen et al., 2025), T-Splade (Lassance, 2023), mSplade (Lassance, 2023). Among these, T-Splade is the approach that translates text into English and encodes the translated text by the Splade model (Formal et al., 2021).

**MILCO configurations.** We consider the following configurations in experiments:

① MILCO (SAP, SCT$_{KD}$, LexEcho): Our strongest setup, which combines alignment with contrastive distillation training and the LexEcho head, producing dual-view lexical representations.

② MILCO (SAP, SCT$_{KD}$, MLM$_{en}$): Similar to (1), but the source view is removed from LexEcho's output, producing only English lexical representations.

③ MILCO (SAP, SCT, LexEcho): Similar to (1), but without distillation. Instead, the InfoNCE loss (Oord et al., 2018) with in-batch negatives is used for Sparse Contrastive Training.

④ MILCO (SAP, MLM$_{en}$): Similar to (2), but without Sparse Contrastive Training.

⑤ MILCO (SCT$_{KD}$, MLM$_{en}$): Similar to (2), but without Sparse Alignment Pre-training.

⑥ noMILCO (SCT$_{KD}$, MLM$_{m}$): A baseline model trained directly with the full multilingual MLM head (without our multilingual connector).

We initialized MILCO from the *bge-m3-unsupervised*[4] multilingual base encoder and initialized the LexEcho head with Splade-v3's English MLM head (Lassance et al., 2024). We use Splade-v3 representations of English text for alignment pretraining. More details on hyperparameters and hardware are provided in Section A.8 of the Appendix.

---

[4]BAAI/bge-m3-unsupervised

## 5 RESULTS AND DISCUSSION

**RQ1: How does MILCO perform compared to state-of-the-art baselines?** Table 1 reports the performance of MILCO and baselines on the MIRACL benchmark (18 languages). Overall, the MILCO ① model, trained with our two-stage pipeline and LexEcho head, achieves the highest effectiveness with an average nDCG@10 of 72.3.

Against sparse baselines, MILCO outperforms M3-Sparse (Chen et al., 2024) by 34.1%, T-Splade (Lassance, 2023) by 32.7%, and MSpladesTok (Lassance, 2023) by 13.1%. Against dense baselines, MILCO still shows substantially higher effectiveness, though with smaller margins. Compared to models of similar size, it outperforms Qwen3 0.6B (Zhang et al., 2025) and M3-Dense (Chen et al., 2024) by 19.5% and 4.5%, respectively, on MIRACL. This advantage generalizes to 39 languages on MTEB v2 cross-lingual and multilingual retrieval (Table 3). Despite being ∼14× smaller, it outperforms E5-Mistral 7B (Wang et al., 2023) and Qwen3 8B on MIRACL, though Qwen3 8B performs better on MTEBv2 where it better leverages task-specific instructions. We additionally train two dense baselines using the same backbone and training data as MILCO. The first, MILCO-dense (align + distill), which uses dense alignment to `thenlper/gte-base`[5] and distillation, achieves an average nDCG@10 of 67.9 on MIRACL. A variant trained with distillation only performs better, reaching 70.9 nDCG@10 on MIRACL. However, both dense baselines still substantially underperform our best sparse MILCO① trained with the two-stage recipe.

Table 2: Performance on Multilingual Long Document Retrieval (nDCG@10, 13 languages). More language-specific details in Table 6.

Table 3: Performance on multilingual and cross-lingual retrieval tasks on Multilingual MTEBv2. (39 languages). More details in Table 7.

| Model | Size | Max Length | Avg |
|---|---|---|---|
| *Dense, multi-vector and hybrid baselines* | | | |
| mE5$_{large}$ | 560M | 512 | 34.2 |
| E5$_{mistral-7b}$ | 7B | 8192 | 42.6 |
| M3-Dense | 560M | 8192 | 52.5 |
| M3-Multi-vector | 560M | 8192 | 57.6 |
| M3-Dense+Sparse | 560M | 8192 | 64.8 |
| M3-All | 560M | 8192 | 65.0 |
| mGTE-TRM Dense | 304M | 8192 | 56.9 |
| mGTE-TRM Dense + Sparse | 304M | 8192 | 71.3 |
| PLAID-X (Multi-vector) | 560M | 512 | 74.2 |
| Qwen3-Embed-0.6B | 0.6B | 32768 | 50.1 |
| Qwen3-Embed-8B | 8B | 32678 | 59.1 |
| *Sparse baselines* | | | |
| BM25 | 2 | 8192 | 53.6 |
| M3-Sparse | 560M | 8192 | 62.2 |
| mGTE-TRM Sparse | 304M | 8192 | 71.0 |
| ① **MILCO** (SAP, SCT$_{KD}$, LexEcho) | 560M | 512 | **74.4** |
| ② **MILCO** (SAP, SCT$_{KD}$, MLM$_{en}$) | 560M | 512 | 69.9 |

| Model | Size | Avg |
|---|---|---|
| *Large Models (≥1B)* | | |
| Qwen3-Embed-8B | 8B | **75.59** |
| jina-embeddings-v4 | 3.8B | 73.84 |
| inf-retriever-v1 | 7.1B | 71.21 |
| SFR-Embedding-Mistral | 7.1B | 68.50 |
| gte-Qwen2-7B-inst | 7B | 67.22 |
| inf-retriever-v1-1.5b | 1.5B | 65.34 |
| gte-Qwen2-1.5B-inst | 1.5B | 65.12 |
| GritLM-7B | 7B | 62.82 |
| NV-Embed-v2 | 7.9B | 58.65 |
| NV-Embed-v1 | 7.9B | 56.64 |
| *Small Models (<1B)* | | |
| gte-multilingual-base | 305M | 64.72 |
| Qwen3-Embed-0.6B | 0.6B | 63.93 |
| bge-m3 | 560M | 62.02 |
| granite-278m-multi | 278M | 55.80 |
| granite-107m-multi | 107M | 49.88 |
| ① **MILCO** (SAP, SCT$_{KD}$, LexEcho) | 560M | 66.83 |

We further evaluate MILCO on the Multilingual Long Document Retrieval (MLDR) benchmark (Table 2). Because MILCO is trained with a 512-token limit, we split long documents into 512-token passages and score documents by their best passage. Under this setup, MILCO achieves an average nDCG@10 of 74.4, which is 14% higher than M3-All, the dense+sparse+multi-vector ensemble, and substantially surpasses Qwen3 0.6B and 8B with native long-context support.

In the Appendices, we report results on LIMIT Test (Weller et al., 2025) (Table 11) and BEIR (Thakur et al., 2021) (Table 10). On LIMIT, MILCO substantially outperforms all dense baselines regardless of size. On BEIR (English), it trails Qwen3 0.6B slightly, but scales better on large collections.

**RQ2: What is the effect of sparse alignment and contrastive training in MILCO?** We observe that Sparse Alignment Pretraining is crucial to ensure that the model's output is grounded in the English vocabulary. In Figure 2, we show two examples of MILCO's output under three training setups. Without SAP, contrastive training leads to semantic collapse, where the model produces

---

[5]`thenlper/gte-base` is similar in size and BEIR (English) performance to our SPLADE-v3 sparse English teacher (GTE-base: 52.61 nDCG@10, SPLADE-v3: 51.69 nDCG@10).

completely random and unexplainable (latent) output tokens with no clear relation to the input. We observe the same effect when we train **noMILCO** ⑥, a multilingual LSR model without the multilingual connector (similar to Splade training). With alignment pretraining, MILCO produces understandable and semantically equivalent English tokens as demonstrated in the figure. However, we observe that both Alignment-only and Contrastive-only result in mediocre multilingual retrieval effectiveness. On MIRACL results in Table 1, Alignment-only **MILCO** ④ and Contrastive-only **MILCO** ⑤ only achieve the average nDCG@10 of 54.5 and 59.2 respectively. Direct training without our connector (noMILCO ⑥) leads to a larger drop in performance, resulting in nDCG@10 of 50.7.

To further improve retrieval effectiveness, we finetune MILCO on retrieval data with a contrastive objective. We experiment with two contrastive losses: InfoNCE with dataset-provided labels (MILCO ③) and KL divergence for knowledge distillation (MILCO ①). With an InfoNCE loss, MILCO ③'s average nDCG@10 improves by 28.62%, from 54.5 with only alignment to 70.1, becoming competitive to BGE-M3-Dense and Multi-vector models. Adding distillation further boosts effectiveness, increasing nDCG@10 to 72.3 and making MILCO subtantially outperform all baselines, including the hybrid BGE-M3 dense-sparse-multivector model and Qwen3 models.

**RQ3: Does the proposed LexEcho head improve robustness?** Unlike dense or multi-vector methods, the transparency of MILCO's sparse, lexicalized representations make errors traceable. When analyzing the English view, we found that representations often miss uncommon entities like *Momo* in Figure 3, leading to reduced retrieval accuracy. In the figure, Doc2 (score = 9.64) is ranked below Doc1 (score = 9.89), despite being more relevant.

The LexEcho head addresses this with a dual-view representation composed of an English view and a source view. When an important entity is missing from the English view, MILCO can fall back to the source view for source-token matching. In Figure 3, LexEcho seems to recognize the model's missing knowledge of *Momo* in English and assigns a high weight to 陌 in the Chinese view. In contrast, for *Apple*, the model relies primarily on English representations (assigning them the highest weights in Doc1 and Doc3) while assigning 苹果 (*Apple*) a low weight in the Chinese view.

On MIRACL (Table 1), MILCO ① with a LexEcho head consistently outperforms MILCO ② with only an English view across all 18 languages, achieving an average nDCG@10 of 72.3 (+4.17% over 69.4). The largest gains occur in non-Latin languages such as Chinese (zh: +8.09%), Telugu (te: +6.8%), Farsi (fa: +6.5%), Korean (ko: +6.5%), and Japanese (ja: +6.04%), where mapping entities into English is particularly difficult since entities could be named differently in English. The benefits of the LexEcho head also extend to long-document retrieval: on MLDR (Table 2), MILCO with LexEcho achieves 74.4 nDCG@10, a 6.43% improvement over MILCO with only an English view (69.5). These highlight the broader robustness of our approach.

**RQ4: Can MILCO perform zero-shot cross-lingual retrieval?** MILCO uses the multilingual connector to maps text across languages into a unified English lexical view. This allows MILCO to perform zero-shot cross-lingual retrieval, which is not possible with sparse models like M3-Sparse that rely on only a source view. We benchmark the cross-lingual capability of MILCO (zero-shot) and baselines on MKQA with R@100 in Table 4.

Prior sparse methods (e.g., BM25, BGE-M3-Sparse) generate source-view representations including input tokens with scalar weights. Their vocabularies are language-specific, so inputs in Chinese

| Alignment Only | Alignment + Contrastive | Contrastive Only |
|---|---|---|
| Input (de): Baltimore Maryland die großartigste Stadt in Amerika (*Baltimore Maryland the greatest city in America*) | | |
| baltimore (2.22), maryland (1.86), city (1.40), greatest (1.25), biggest (0.98), garrison (0.35), geography (0.31), tourism (0.19) … | baltimore (1.77), city (1.52), maryland (1.23), america (1.19), greatest (0.89), usa (0.81), best (0.62), urban (0.26) … | governing (1.17), past (1.07), match (0.95), worn (0.86), sky (0.65), gas (0.52), boot (0.34), mayor (0.31) … |
| Input (vi): Giá trị tài sản ròng của Tesla là bao nhiêu? (*What is Tesla's net worth?*) | | |
| tesla (3.36), worth (2.61), price (1.84), net (1.52), salary (1.14), money (0.80), mining (0.35), generation (0.33) … | tesla (3.02), worth (1.89), price (1.47), net (1.37), wealth (0.71), asset (0.63), stock (0.42), company (0.41) … | relative (0.97), drinks (0.75), gaelic (0.75), contaminated (0.73), sigh (0.67), webb (0.60), dust (0.46), – (1.22) … |

Figure 2: Sparse representations with different training strategies. *Alignment only* produces many grounded tokens (**green**) but also distantly relevant tokens (**orange**), *Contrastive* further prunes and refines. *Contrastive-only* suffers from semantic collapse, drifting toward ungrounded tokens (**red**).

| Input | Translation | (1) LexEcho (English View) | (2) LexEcho (Source View) | Score (1) | Score (1)+(2) |
|---|---|---|---|---|---|
| **Query:** 陌陌 直播音乐怎么导入手机? | How to import **Momo** Live Music into mobile phone? | music(1.16), import(0.95), phone(0.88), step(0.80), transfer(0.72), no(0.69), songs(0.55), live(0.51), song(0.49), phones(0.46), stream(0.44) ... | 陌(1.46), 手机(0.97), 导(0.68), 直播(0.67), 音乐(0.65), (0.38), 怎么(0.38), 入(0.36), ?(0.34), _(0.19) | | |
| **Doc1:**用户可将苹果音乐歌曲下载或录音保存，再导入手机播放 | Users can download **Apple** Music songs and then import them into their phones … | apple(1.72), music(1.58), step(1.33), songs(1.30), download(1.27), phone(1.18), is(1.18), play(1.14), song(1.09), can(1.03), save(1.03), app(0.95),... | 音乐(0.46), 用户(0.45), 歌曲(0.44), 选择(0.40), 导(0.40), 的方式(0.34), 上的(0.34), 保存(0.28), 可以(0.27), 苹果(0.23) … | **9.89** Rank 1 | **11.66** Rank 2 |
| **Doc2:** 陌陌 直播的歌曲可以用保存功能 转到手机. | Songs from **Momo** Live can be saved to your phone using the save function. | step(1.62), songs(1.32), save(1.29), phone(1.20), song(1.15), music(1.07), storage(1.01), live(0.84), can(0.81), transfer(0.81), stream(0.79), … | 陌(1.54), 歌曲(0.72), 直播(0.71), 功能(0.67), 手机(0.64), 你可以(0.59), 保存(0.56), 用(0.54), _(0.53), 把(0.51) ... | **9.64** Rank 2 | **13.27** Rank 1 |

Figure 3: The tail entity *Momo* is missing in the English view of the query and Doc2, reducing Doc2's score despite its higher relevance. The LexEcho head resolves this by selectively retaining missing entities from source tokens, correctly ranking Doc2 on top.

yield only Chinese tokens. This causes vocabulary mismatch and poor cross-lingual retrieval, with average R@100 scores of just 39.9 and 45.3 on MKQA. In contrast, MILCO avoids this issue with a shared English lexical space. Despite not being trained for cross-lingual retrieval, MILCO achieves strong results on MKQA, with a zero-shot R@100 of 76.6, improving 91.9% and 69.1% over BM25 and BGE-M3-Sparse, respectively.

Dense and multi-vector methods operate in a latent space, so they do not suffer from vocabulary mismatch and perform reasonably well on MKQA. BGE-Dense and multi-vector models are among the strongest baselines, with an average R@100 of around 75. While these methods outperform sparse baselines (e.g., BM25 or BGE-M3-Sparse), MILCO achieves about 1.7–1.9% higher R@100 than the BGE dense and multi-vector baselines, while also retaining the transparency that facilitate model analysis and error tracing. MILCO is about 9% and 12.8% better than E5-Mistral 7B and Qwen3 8B, respectively, despite having only 560M parameters.

Table 4: Cross-lingual retrieval performance on MKQA, averaged across 25 languages. More details in Table 9.

| Model | Avg (R@100) |
|---|---|
| **Baselines** | |
| E5-large | 70.9 |
| E5-mistral-7b | 70.1 |
| BGE-M3 Dense | 75.1 |
| BGE-M3 Multi-Vec | 75.3 |
| BGE-M3 Dense+Sparse | 75.3 |
| PLAID-X (multivector) | 73.4 |
| Qwen3-Embed-0.6B | 54.4 |
| Qwen3-Embed-8B | 67.9 |
| BM25 | 39.9 |
| BGE-M3 Sparse | 45.3 |
| ① **MILCO** (SAP, SCT$_{KD}$, LexEcho) | **76.6** |

## 6 Efficiency and Effectiveness Tradeoffs

**Model Size vs. Effectiveness.** In Figure 4, we plot MILCO's effectiveness against model size compared to baselines. We observe that MILCO, with 560M parameters, is the most effective model within its size range and even substantially outperforms larger models (e.g., Qwen3-8B and E5-Mistral-7B) across all 18 MIRACL languages. With the same model size, the BGE-M3-Multivector model underperforms MILCO despite producing multiple dense vectors for each query/document.

**Sparsity vs. Effectiveness.** Dense retrieval models like Matryoshka representations (Kusupati et al., 2022) support truncating embeddings for efficiency, but require additional Matryoshka training losses. MILCO and LSR methods naturally allow post-hoc pruning (Lei et al., 2025; Wen et al., 2025; Bruch et al., 2024), because LSR encodes queries and documents as weighted tokens that can be ranked and truncated at inference. Unlike Matryoshka, which applies the same truncation size to all inputs, LSR supports variable $k$ (e.g., fewer tokens for shorter texts). Figure 5 compares two pruning strategies: top-$k$ pruning and mass-based pruning, which removes the p-tail percentile of token weights, yielding variable tokens per document. Exact numbers are included in the Appendix 12. Mass-based pruning delivers a slightly more favorable trade-off than top-k pruning. At $p = 95$, it averages only 30 tokens/document yet already surpasses Qwen3-Embed 0.6B on nDCG@10 (62.2). It reaches 96% of full performance at $p = 86$ (86.4 tokens/doc) and achieves SOTA at 300 tokens, with only marginal gains beyond. With LexEcho's vocabulary of 280k terms, activating just 300 tokens (0.1%) yields representations that are 99.9% sparse, transparent, and highly effective.

**Sparsity vs. Efficiency.** We report index statistics and retrieval latency with MILCO under an inverted-index setting. We use Seismic (Bruch et al., 2024), an ANN method built on top of an inverted index. All results are on MIRACL (English subset, about 32M passages), with retrieval run

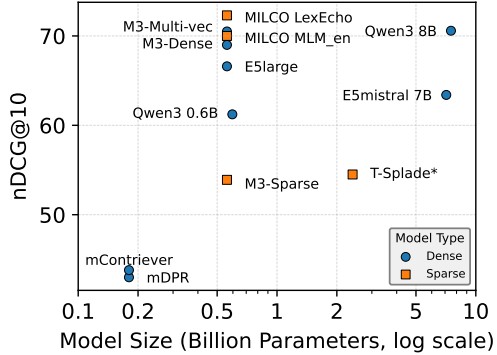

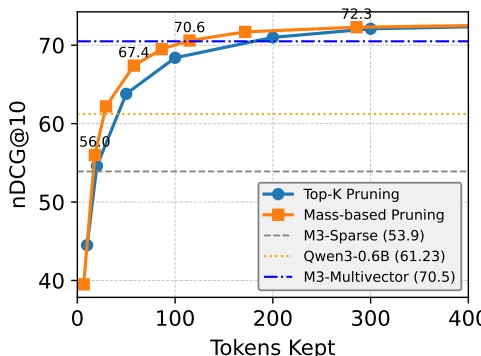

Figure 4: Model size versus effectiveness on MIRACL. MILCO is lightweight (560M params), while being highly effective.

Figure 5: Effectiveness (nDCG@10, MIRACL) of MILCO with varying sparsity levels obtained by post-hoc pruning methods.

Table 5: Retrieval Latency of MILCO sparse retrieval with Seismic and Qwen3-Embedding-0.6B dense retrieval with Faiss.

| Model | Index | Avg Latency (ms) | P95 Latency (ms) | QPS | nDCG@10 | Index Size |
|---|---|---|---|---|---|---|
| Qwen3-Embed-0.6B | HNSW | 1.29 | 1.47 | 777 | 50.4 | 134G |
| MILCO (p=0) | Seismic | 1.85 | 4.32 | 538 | 56.4 | 61G |
| MILCO (p=10) | Seismic | 1.29 | 2.92 | 774 | 56.3 | 40G |
| MILCO (p=30) | Seismic | 0.61 | 1.26 | 1647 | 54.4 | 25G |
| MILCO (p=50) | Seismic | 0.65 | 1.33 | 1545 | 53.3 | 16G |
| MILCO (p=60) | Seismic | 0.44 | 0.82 | 2265 | 50.3 | 12G |

on a single AMD EPYC 7763 CPU core. We build several indexes of the pivot view with hyperparameters (n_postings=15000, query_cut=10, heap_factor=0.9) and different amounts of post-hoc pruning. The unpruned index ($p = 0$) has an average posting-list length of 4636.09, resulting in a 61 GB index and an average retrieval latency of 1.85 ms/query. We then prune the lowest-weight dimensions whose cumulative weights account for $p \in \{10, 30, 50, 60\}\%$ of the total, which shrinks the inverted index and speeds up retrieval. Results are shown in Table 5.

Regarding index size, post-hoc pruning substantially shrinks the index: 40 GB at $p = 10$, 25 GB at $p = 30$, 16 GB at $p = 50$, and 12 GB at $p = 60$. Thus, at the most aggressive pruning level, we reduce index size by roughly 80% while keeping strong retrieval effectiveness. To contextualize these index sizes, we compare against Qwen3-Embedding-0.6B with a Faiss dense HNSW index (Douze et al., 2024) ($M = 32$, ef $= 64$) on the same hardware and collection. This dense baseline has an index size of 134 GB, whereas MILCO's pruned Seismic index is already smaller at $p = 10$ (40 GB) and becomes 5–10× smaller at higher pruning levels (25 GB at $p = 30$, 12 GB at $p = 60$).

Regarding latency, pruning also yields consistent improvements over the full representation: $p = 10$ reduces average latency from 1.85 ms to 1.29 ms (∼30% speed-up) with virtually no loss in nDCG@10 (56.4 → 56.3), $p = 30$ makes queries about 3× faster (0.61 ms, nDCG@10 = 54.4), and even $p = 60$ achieves a $> 4\times$ speed-up (0.44 ms) while remaining competitive (nDCG@10 = 50.3). For comparison, the Qwen3-Embedding-0.6B + Faiss HNSW achieves an average latency of 1.29 ms and nDCG@10 = 50.4. Seismic's ANN inverted index with pruning therefore allows MILCO to (i) match this latency at $p = 10$ while achieving substantially higher effectiveness (nDCG@10 = 56.3), and (ii) further reduce latency to $\approx 0.44$ ms at $p = 60$, while remaining at least as effective overall.

## 7 CONCLUSION

We introduced MILCO, a novel multilingual learned sparse retriever that connects 39 languages to a shared English lexical space through a lightweight connector. Alignment pretraining enables the use of contrastive training, whereas the LexEcho preserves entities lost during cross-lingual projection. MILCO delivers strong zero-shot cross-lingual retrieval, showing competitive performance without cross-lingual retrieval training. Overall, MILCO achieves state-of-the-art multilingual retrieval results, while offering transparent representations and efficient post-hoc pruning.

## REPRODUCIBILITY STATEMENT

We have taken several measures to ensure the transparency and reproducibility of our work.

**Datasets.** All datasets used for pretraining and training MILCO models are publicly available. Table 14 (Appendix) reports statistics on the number and sources of parallel sentences used for Sparse Alignment Pretraining, while Table 15 (Appendix) describes the datasets used for Sparse Contrastive Training. These datasets are widely adopted in prior work on dense retrieval (Wang et al., 2024; Chen et al., 2024; Li et al.). We note, however, that some recent models such as Qwen3-Embed (Zhang et al., 2025) do not disclose their training data, which makes direct comparisons not strictly fair.

**Hyper-parameters and hardware.** All hyper-parameters and hardware specifications used for Sparse Alignment and Sparse Contrastive Training are described in Section A.8 (Appendix). Any hyper-parameters not explicitly listed are set to the default values provided in HuggingFace's Trainer (Wolf et al., 2019), which we use to train our models. During pretraining and training, we hard-coded the random seed to 42.

**Code.** Our code is available at: `https://github.com/thongnt99/milco`.

**Models and evaluation.** MILCO is trained on 63 languages and evaluated on 39 languages, as detailed in Section A.9 (Appendix). Trained MILCO checkpoints are released at: `https://github.com/thongnt99/milco`. To illustrate the multilingual capabilities of our model, we provide example inputs and their corresponding sparse representations across multiple languages in Section A.1 (Appendix).

## ETHICS STATEMENT

We present MILCO, a multilingual learned sparse retrieval method supporting 39 languages. All datasets and models used to train MILCO are publicly available, and we do not introduce any proprietary or sensitive data. Since our work builds on public data, it may reflect biases present in those sources. We aims to broaden access to multilingual information retrieval research, especially for underrepresented languages.

## ACKNOWLEDGMENT

This research was supported by the Hybrid Intelligence Center, a 10-year program funded by the Dutch Ministry of Education, Culture and Science through the Netherlands Organisation for Scientific Research, project VI.Vidi.223.166 of the NWO Talent Programme which is (partly) financed by the Dutch Research Council (NWO). We acknowledge the Dutch Research Council for awarding this project access to the LUMI supercomputer, owned by the EuroHPC Joint Undertaking, hosted by CSC (Finland) and the LUMI consortium through project number NWO-2024.050.

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

# A  APPENDIX

## A.1  DEMONSTRATION EXAMPLES

In Figure 6, we present a list of demonstration examples. The inputs are in different languages, while the outputs are bag-of-words English tokens produced by our ① MILCO (SAP, SCT$_{KD}$, LexEcho) model. These examples illustrate that MILCO generates transparent representations, making it possible for humans to interpret, inspect, and trace potential errors or biases.

| ID | Language | Input Text | Sparse Representation (English View) |
|----|----------|-----------|--------------------------------------|
| 1 | en | what is tesla net worth? | {"tesla": 3.4, "worth": 2.72, "net": 1.7, "price": 1.65, "salary": 0.8, "nikola": 0.78, "stock": 0.58, "electric": 0.52, "car": 0.49, "generation": 0.49, "money": 0.48, "levi": 0.42, "mining": 0.31, "company": 0.27, "sale": 0.2, "milan": 0.12, "edmund": 0.12, "revenue": 0.06} |
| 1 | hi | टेस्ला की कुल संपत्ति क्या है? | {"tesla": 3.35, "worth": 2.79, "net": 1.73, "price": 1.56, "salary": 1.03, "money": 0.78, "generation": 0.51, "stock": 0.5, "nikola": 0.49, "electric": 0.4, "car": 0.31, "company": 0.29, "mining": 0.29, "levi": 0.25, "wealth": 0.17, "total": 0.16, "sale": 0.11, "milan": 0.1} |
| 1 | zh | 特斯拉的净资产是多少？ | {"tesla": 3.35, "worth": 2.71, "price": 1.91, "net": 1.76, "salary": 1.35, "money": 0.95, "stock": 0.79, "company": 0.58, "electric": 0.54, "nikola": 0.54, "generation": 0.49, "car": 0.34, "mining": 0.32, "levi": 0.19, "motors": 0.15, "total": 0.03} |
| 1 | de | Was ist Teslas Nettowert? | {"tesla": 3.34, "worth": 2.69, "net": 1.94, "price": 1.51, "stock": 0.78, "salary": 0.68, "money": 0.62, "electric": 0.48, "nikola": 0.46, "generation": 0.43, "sale": 0.34, "car": 0.25, "levi": 0.22, "company": 0.19, "mining": 0.16, "milan": 0.11, "wealth": 0.11, "edmund": 0.11, "currently": 0.1, "revenue": 0.1} |
| 1 | nl | Wat is de nettowaarde van Tesla? | {"tesla": 3.36, "worth": 2.75, "net": 1.89, "price": 1.66, "salary": 0.99, "stock": 0.87, "money": 0.61, "electric": 0.55, "nikola": 0.49, "company": 0.41, "generation": 0.41, "sale": 0.32, "levi": 0.27, "mining": 0.23, "car": 0.22, "revenue": 0.15, "edmund": 0.12, "currently": 0.11, "wealth": 0.08, "investment": 0.06} |
| 1 | ar | ما هي القيمة الصافية لشركة تيسلا؟ | {"tesla": 3.31, "worth": 2.64, "net": 1.81, "price": 1.52, "salary": 0.85, "stock": 0.82, "money": 0.58, "company": 0.53, "electric": 0.43, "levi": 0.43, "generation": 0.42, "nikola": 0.28, "mining": 0.26, "revenue": 0.22, "sale": 0.18, "car": 0.16, "total": 0.1, "investment": 0.08, "currently": 0.03, "wealth": 0.03} |
| 1 | vi | Giá trị tài sản ròng của Tesla là bao nhiêu? | {"tesla": 3.36, "worth": 2.61, "price": 1.84, "net": 1.52, "salary": 1.14, "money": 0.8, "stock": 0.7, "electric": 0.52, "nikola": 0.45, "mining": 0.35, "generation": 0.33, "sale": 0.27, "company": 0.25, "car": 0.16, "wealth": 0.16, "investment": 0.14, "levi": 0.09, "revenue": 0.08, "total": 0.05} |
| 2 | en | Baltimore Maryland the greatest city in America | {"baltimore": 2.47, "maryland": 2.07, "greatest": 1.68, "city": 1.59, "biggest": 1.38, "md": 1.26, "america": 1.11, "usa": 1.09, "great": 0.98, "cities": 0.93, "town": 0.7, "american": 0.47, "us": 0.45, "urban": 0.41, "birthplace": 0.41, "was": 0.34, "famous": 0.29, "garrison": 0.25, "beautiful": 0.23, "geography": 0.21} |
| 2 | hi | बाल्टीमोर मैरीलैंड अमेरिका का सबसे बड़ा शहर | {"baltimore": 2.63, "maryland": 2.08, "largest": 1.7, "city": 1.55, "biggest": 1.5, "md": 1.49, "usa": 1.14, "cities": 1.06, "us": 0.78, "town": 0.65, "america": 0.65, "population": 0.64, "urban": 0.48, "metropolitan": 0.34, "census": 0.31, "garrison": 0.3, "geography": 0.29, "size": 0.26, "headquarters": 0.26, "big": 0.19} |
| 2 | zh | 马里兰州巴尔的摩是美国最伟大的城市 | {"baltimore": 2.66, "maryland": 2.07, "greatest": 1.85, "city": 1.73, "md": 1.65, "biggest": 1.53, "usa": 1.38, "great": 1.29, "cities": 1.26, "us": 1.02, "america": 0.91, "town": 0.81, "best": 0.66, "urban": 0.61, "geography": 0.57, "birthplace": 0.57, "beautiful": 0.55, "garrison": 0.51, "headquarters": 0.47, "location": 0.46} |
| 2 | de | Baltimore Maryland die großartigste Stadt in Amerika | {"baltimore": 2.22, "maryland": 1.86, "city": 1.4, "greatest": 1.25, "md": 1.16, "biggest": 0.98, "great": 0.96, "cities": 0.95, "usa": 0.94, "america": 0.91, "town": 0.69, "beautiful": 0.57, "most": 0.52, "american": 0.49, "us": 0.46, "urban": 0.43, "best": 0.37, "garrison": 0.35, "geography": 0.31, "birthplace": 0.29} |
| 2 | nl | Baltimore Maryland de beste stad in Amerika | {"baltimore": 2.35, "maryland": 2.0, "best": 1.94, "city": 1.51, "md": 1.27, "usa": 1.17, "america": 1.11, "beautiful": 1.05, "cities": 1.02, "biggest": 0.89, "town": 0.77, "american": 0.61, "us": 0.54, "urban": 0.49, "birthplace": 0.37, "garrison": 0.35, "popular": 0.3, "tourism": 0.29, "location": 0.28, "headquarters": 0.27} |
| 2 | ar | بالتيمور ماريلاند أعظم مدينة في أمريكا | {"baltimore": 2.57, "maryland": 2.09, "city": 1.63, "greatest": 1.55, "md": 1.53, "biggest": 1.36, "great": 1.24, "usa": 1.21, "cities": 1.15, "america": 1.14, "town": 0.72, "garrison": 0.66, "us": 0.61, "location": 0.57, "urban": 0.57, "headquarters": 0.53, "american": 0.52, "best": 0.47, "beautiful": 0.43, "birthplace": 0.42} |
| 2 | vi | Baltimore Maryland thành phố vĩ đại nhất ở Mỹ | {"baltimore": 2.45, "maryland": 2.05, "city": 1.47, "biggest": 1.38, "md": 1.28, "greatest": 1.08, "usa": 1.05, "cities": 0.95, "largest": 0.92, "us": 0.71, "america": 0.69, "town": 0.65, "great": 0.47, "urban": 0.42, "headquarters": 0.34, "metropolitan": 0.3, "garrison": 0.29, "population": 0.27, "metropolis": 0.22, "american": 0.21} |
| 3 | zh | 有谁知道陌陌直播的音乐怎么导入手机里 | {"import": 1.6, "music": 1.46, "live": 1.17, "phone": 1.16, "imported": 0.95, "songs": 0.86, "app": 0.84, "step": 0.71, "youtube": 0.68, "phones": 0.65, "stream": 0.63, "no": 0.62, "button": 0.62, "download": 0.58, "song": 0.48, "pandora": 0.48, "mp3": 0.47, "player": 0.46, "sync": 0.46, "transfer": 0.45} |
| 4 | de | Alan Smithee is a pseudonym for a fictional director responsible for films in which the actual director does not want his name associated with the work. From 1968 to 2000, it was recommended by the Directors Guild of America (DGA) for such situations. | {"alan": 3.02, "##ee": 3.0, "smith": 2.95, "pseudonym": 2.15, "director": 2.05, "directors": 1.93, "fictional": 1.9, "##ga": 1.86, "guild": 1.85, "responsible": 1.61, "d": 1.58, "who": 1.49, "film": 1.34, "alias": 1.29, "directed": 1.26, "##ees": 1.2, "actual": 1.09, "1968": 1.07, "recommended": 1.05, "directing": 1.04} |
| 5 | fr | Paul Jules Antoine Meillet, né le à Moulins (Allier) et mort le à Châteaumeillant (Cher), est le principal linguiste français des premières décennies du . Il est aussi philologue. | {"mei": 2.9, "##llet": 2.73, "jules": 2.62, "paul": 2.57, "antoine": 2.46, "##ulin": 2.41, "allie": 2.3, "linguist": 2.21, "french": 2.08, "cher": 2.04, "chateau": 1.84, "france": 1.77, "##mei": 1.72, "mo": 1.7, "who": 1.69, "linguistics": 1.66, "##llan": 1.6, "born": 1.51, "died": 1.39, "##let": 1.34} |
| 6 | zh | 金章宗完颜璟 女真名麻達葛，金朝第6位皇帝（1189年1月20日—1208年12月29日在位），在位19年，享年41岁。章宗為金世宗完颜雍之嫡孙，其在位期間修訂國內律法，政治清明，史稱明昌之治。章宗統治下的金朝文化發展達至頂峰，但同時軍事能力卻也日益低下，蒙古帝國也於同時崛起 | {"dynasty": 2.19, "emperor": 2.03, "qing": 2.01, "ming": 1.96, "sima": 1.8, "kim": 1.69, "sixth": 1.67, "mongolian": 1.64, "6": 1.61, "khan": 1.54, "korea": 1.45, "age": 1.44, "##chang": 1.44, "dynasties": 1.42, "date": 1.36, "who": 1.33, "died": 1.32, "china": 1.31, "reign": 1.31, "empire": 1.3} |

Figure 6: Examples of MILCO's output representations (English view) on different languages.

## A.2 DETAILED MULTILINGUAL/CROSS-LINGUAL RETRIEVAL RESULTS

In this section, we show the detailed language-specific results of MILCO and baselines in the following multilingual and cross-lingual retrieval datasets. The result on Multilingual Long Document Retrieval (MLDR) is shown in Table 6. The result on MTEBv2 (multilingual and cross-lingual tasks) is shown in Table 7. The result on the NeuCLIR benchmark (cross-lingual retrieval) is shown in Table 8. The result on MKQA (cross-lingual retrieval) is shown in Table 9.

Table 6: Multilingual (long) document retrieval on the MLDR (measured by nDCG@10).

| Model | Max Length | Avg | ar | de | en | es | fr | hi | it | ja | ko | pt | ru | th | zh |
|---|---|---|---|---|---|---|---|---|---|---|---|---|---|---|---|
| *Dense and multi-vector baselines* | | | | | | | | | | | | | | | |
| mE5$_{large}$ | 512 | 34.2 | 33.0 | 26.9 | 33.0 | 51.1 | 49.5 | 21.0 | 43.1 | 29.9 | 27.1 | 58.7 | 42.4 | 15.9 | 13.2 |
| E5$_{mistral-7b}$ | 8192 | 42.6 | 29.6 | 40.6 | 43.3 | 70.2 | 60.5 | 23.2 | 55.3 | 41.6 | 32.7 | 69.5 | 52.4 | 18.2 | 16.8 |
| M3-Dense | 8192 | 52.5 | 47.6 | 46.1 | 48.9 | 74.8 | 73.8 | 40.7 | 62.7 | 50.9 | 42.9 | 74.4 | 59.5 | 33.6 | 26.0 |
| M3-Multi-vector | 8192 | 57.6 | 56.6 | 50.4 | 55.8 | 79.5 | 77.2 | 46.6 | 66.6 | 52.8 | 48.8 | 77.5 | 64.2 | 39.4 | 32.7 |
| M3-Dense+Sparse | 8192 | 64.8 | 63.0 | 56.4 | 64.2 | 88.7 | 84.2 | 52.3 | 75.8 | 58.5 | 53.1 | 86.0 | 75.6 | 42.9 | 42.0 |
| M3-All | 8192 | 65.0 | 64.7 | 57.9 | 63.8 | 86.8 | 83.9 | 52.2 | 75.5 | 60.1 | 55.7 | 85.4 | 73.8 | 44.7 | 40.0 |
| PLAID-X (Multi-vector) | 512 | 74.2 | 78.5 | 65.5 | 81.4 | 90.9 | 87.5 | 64.0 | 84.2 | 67.3 | 66.9 | 85.5 | 86.9 | 43.7 | 62.7 |
| Qwen3-Embed - 0.6B | 32768 | 50.1 | 44.7 | 45.0 | 75.5 | 48.4 | 69.7 | 24.8 | 62.6 | 49.7 | 38.3 | 73.2 | 61.2 | 30.7 | 26.9 |
| Qwen3-Embed - 8B | 32678 | 59.1 | 57.7 | 54.5 | 86.1 | 56.1 | 79.5 | 35.1 | 72.7 | 58.3 | 50.4 | 79.6 | 69.6 | 37.9 | 30.8 |
| *Sparse baselines* | | | | | | | | | | | | | | | |
| BM25 | 8192 | 53.6 | 45.1 | 52.6 | 57.0 | 78.0 | 75.7 | 43.7 | 70.9 | 36.2 | 25.7 | 82.6 | 61.3 | 33.6 | 34.6 |
| M3-Sparse | 8192 | 62.2 | 58.7 | 53.0 | 62.1 | 87.4 | 82.7 | 49.6 | 74.7 | 53.9 | 47.9 | 85.2 | 72.9 | 40.3 | 40.5 |
| ① MILCO (SAP, SCT$_{KD}$, LexEcho) | 512 | **74.4** | **75.3** | **66.1** | **82.5** | **93.2** | **90.8** | **59.5** | **81.9** | **68.8** | **67.9** | **90.7** | **85.5** | **45.8** | **59.0** |

Table 7: Performance of embedding models on MTEBv2's multilingual and cross-lingual retrieval tasks (Enevoldsen et al., 2025). MILCO outperforms other models with similar sizes (e.g, Qwen3-0.6B, BGE-M3), while under-performs larger models, such as Qwen3-Embed-8B. (M = Multilingual, C = Cross-lingual). We evaluate English-only retrieval tasks separately in Section A.3.

| Model | Avg. | Belebele (M) | MIRACL-HN (M) | MLQA (C) | Statcan (M) | Twitter (M) | Wiki (C) |
|---|---|---|---|---|---|---|---|
| gte-multilingual-base | 64.72 | 77.60 | 64.17 | 72.19 | 21.74 | 68.92 | 83.69 |
| bge-m3 | 62.02 | 78.16 | 69.59 | 74.81 | 21.86 | 37.82 | 89.87 |
| granite-278m-multi | 55.80 | 62.20 | 59.45 | 62.99 | 30.14 | 34.98 | 85.06 |
| granite-125m-eng | 26.99 | 33.37 | 16.35 | 22.90 | 30.71 | 5.92 | 52.70 |
| granite-107m-multi | 49.88 | 55.12 | 57.25 | 60.47 | 27.50 | 17.06 | 81.88 |
| gte-Qwen2-7B-inst | 67.22 | 77.54 | 51.58 | 78.69 | 37.87 | 68.64 | 88.97 |
| gte-Qwen2-1.5B-inst | 65.12 | 66.59 | 63.23 | 72.89 | 33.25 | 67.01 | 87.77 |
| NV-Embed-v2 | 58.65 | 69.79 | 55.54 | 70.61 | 19.55 | 45.57 | 90.83 |
| inf-retriever-v1 | 71.21 | 77.37 | 60.93 | 80.31 | 37.30 | 79.30 | 92.02 |
| jina-embeddings-v4 | 73.84 | 74.29 | 62.95 | 74.90 | 58.07 | 84.38 | 88.46 |
| inf-retriever-v1-1.5b | 65.34 | 66.06 | 62.35 | 72.93 | 31.31 | 70.46 | 88.93 |
| Qwen3-Embed-0.6B | 63.93 | 68.74 | 61.23 | 72.79 | 33.63 | 60.04 | 87.13 |
| Qwen3-Embed-8B | 75.59 | 88.81 | 70.58 | 83.55 | 40.46 | 78.20 | 91.96 |
| ① MILCO (SAP, SCT$_{KD}$, LexEcho) | 66.83 | 80.72 | 72.65 | 83.00 | 24.00 | 50.00 | 90.63 |

Table 8: Results on NeuCLIR cross-lingual benchmarks (Lawrie et al., 2024) on three languages (Chinese, Persian, and Russian). The Avg. MLIR score (nDCG@20) is the mean across the two years. Our 560M MILCO model outperforms Qwen3 0.6B, but falls behind PLAID-X and Qwen3-Embed 4B/8B. PLAID-X focuses exclusively on the test languages.

| Model | 2023 MLIR (C) | 2024 MLIR (C) | Avg. MLIR (C) |
|---|---|---|---|
| SPLADE v3 (transl. docs) | 0.420 | 0.440 | 0.430 |
| PLAID-X | 0.404 | 0.468 | 0.436 |
| Qwen3-Embed 0.6B | 0.317 | 0.311 | 0.314 |
| Qwen3-Embed 4B | 0.440 | 0.415 | 0.428 |
| Qwen3-Embed 8B | 0.434 | 0.419 | 0.427 |
| ① MILCO (SAP, SCT$_{KD}$, LexEcho) | 0.395 | 0.427 | 0.411 |

Table 9: Cross-lingual retrieval performance on MKQA (Recall@100). Abbreviations: mCtr = mContriever, OA3 = OpenAI-3, PLD = PLAID-X, Q0.6 = Qwen3-0.6B, Q8 = Qwen3-8B.

| Lang | Dense Baselines | | | | | | | | | | | | Sparse Baselines | | *MILCO* |
|------|------|------|------|--------|------|------|-------|-------|--------|------|------|------|------|------|------|
| | mDPR | mCtr | E5-L | E5-M7B | OA3 | M3-D | M3-MV | M3-DS | M3-All | PLD | Q0.6 | Q8 | BM25 | M3-S | |
| ar | 48.2 | 58.2 | 68.7 | 59.6 | 65.6 | 71.1 | 71.4 | 71.1 | 71.5 | 64.2 | 44.8 | 64.75 | 18.9 | 23.5 | 74.9 |
| da | 67.4 | 73.9 | 77.4 | 77.8 | 73.6 | 77.2 | 77.5 | 77.4 | 77.6 | 77.0 | 57.9 | 69.89 | 49.3 | 55.4 | 77.9 |
| de | 65.8 | 71.7 | 76.9 | 77.0 | 73.6 | 76.2 | 76.3 | 76.4 | 76.3 | 76.0 | 61.6 | 69.69 | 35.4 | 43.3 | 76.6 |
| es | 66.8 | 72.6 | 76.6 | 77.4 | 73.9 | 76.4 | 76.6 | 76.7 | 76.9 | 75.7 | 62.5 | 70.75 | 43.4 | 50.6 | 77.7 |
| fi | 56.2 | 70.2 | 74.0 | 72.0 | 72.7 | 75.1 | 75.3 | 75.7 | 75.6 | 70.5 | 46.5 | 65.06 | 46.3 | 51.1 | 76.6 |
| fr | 68.2 | 73.8 | 76.5 | 77.0 | 76.2 | 76.2 | 76.4 | 76.6 | 76.6 | 76.1 | 61.8 | 70.22 | 45.3 | 53.9 | 77.4 |
| he | 49.7 | 63.2 | 69.0 | 67.2 | 58.1 | 72.4 | 72.9 | 72.5 | 73.0 | 70.5 | 39.0 | 62.68 | 26.9 | 31.1 | 74.8 |
| hu | 60.4 | 69.7 | 74.7 | 75.0 | 71.2 | 74.7 | 74.6 | 74.9 | 75.0 | 72.2 | 43.7 | 65.07 | 38.2 | 44.6 | 75.8 |
| it | 66.0 | 72.3 | 76.8 | 77.1 | 73.6 | 76.0 | 76.4 | 76.3 | 76.5 | 75.2 | 61.3 | 69.98 | 45.2 | 52.5 | 77.5 |
| ja | 60.3 | 64.8 | 71.5 | 65.1 | 71.9 | 75.0 | 75.1 | 75.0 | 75.2 | 75.2 | 57.2 | 69.45 | 24.5 | 31.3 | 77.2 |
| km | 29.5 | 26.8 | 33.4 | 34.3 | 33.9 | 68.6 | 69.1 | 68.8 | 69.2 | 63.7 | 24.6 | 52.76 | 20.6 | 30.1 | 70.4 |
| ko | 50.9 | 59.7 | 68.1 | 59.4 | 73.3 | 71.6 | 71.7 | 71.6 | 71.8 | 70.7 | 47.2 | 65.51 | 27.9 | 31.4 | 73.9 |
| ms | 65.5 | 74.1 | 76.3 | 77.0 | 73.3 | 77.2 | 77.4 | 77.4 | 77.4 | 75.5 | 59.8 | 70.06 | 55.9 | 62.4 | 78.0 |
| nl | 68.2 | 73.7 | 77.0 | 79.1 | 74.2 | 77.2 | 77.7 | 77.7 | 77.6 | 76.5 | 58.6 | 71.23 | 56.2 | 62.4 | 78.3 |
| no | 66.7 | 73.5 | 77.3 | 76.6 | 73.3 | 77.1 | 77.2 | 77.4 | 77.4 | 76.3 | 55.9 | 69.09 | 52.1 | 57.9 | 77.6 |
| pl | 67.0 | 71.5 | 73.0 | 77.1 | 73.3 | 76.3 | 76.5 | 76.3 | 76.4 | 75.1 | 54.7 | 68.86 | 48.0 | 50.5 | 76.9 |
| pt | 65.5 | 72.6 | 73.5 | 77.5 | 73.7 | 76.3 | 76.4 | 76.5 | 76.4 | 74.4 | 61.0 | 69.97 | 44.9 | 50.9 | 77.4 |
| ru | 62.7 | 69.8 | 76.8 | 75.5 | 72.0 | 76.2 | 76.4 | 76.2 | 76.5 | 76.2 | 58.9 | 69.65 | 33.2 | 36.9 | 77.4 |
| sv | 66.9 | 73.2 | 77.6 | 78.3 | 74.0 | 76.9 | 77.2 | 77.4 | 77.4 | 76.5 | 55.8 | 69.71 | 54.6 | 59.6 | 78.2 |
| th | 53.8 | 66.9 | 76.0 | 67.4 | 65.2 | 75.6 | 75.9 | 76.0 | 76.6 | 76.2 | 56.3 | 69.72 | 37.8 | 45.0 | 77.9 |
| tr | 59.1 | 71.1 | 74.3 | 74.9 | 75.2 | 75.6 | 75.9 | 76.0 | 76.0 | 72.0 | 52.1 | 66.57 | 45.8 | 51.8 | 77.6 |
| vi | 63.4 | 70.9 | 75.4 | 77.0 | 71.1 | 76.6 | 76.7 | 76.8 | 76.9 | 74.3 | 57.6 | 69.09 | 46.6 | 51.8 | 77.9 |
| zh_cn | 63.7 | 68.1 | 56.6 | 69.3 | 70.7 | 74.6 | 74.9 | 74.7 | 75.0 | 72.7 | 62.1 | 69.51 | 31.0 | 35.4 | 76.1 |
| zh_hk | 62.8 | 68.0 | 58.4 | 65.1 | 69.6 | 73.8 | 74.1 | 74.0 | 74.3 | 72.1 | 58.9 | 68.39 | 35.0 | 39.8 | 75.3 |
| zh_tw | 64.0 | 67.9 | 58.1 | 68.5 | 69.6 | 73.5 | 73.5 | 73.6 | 73.6 | 71.4 | 59.2 | 69.13 | 33.5 | 37.7 | 75.5 |
| **Avg** | 60.6 | 67.9 | 70.9 | 70.1 | 69.5 | 75.1 | 75.3 | 75.3 | 75.5 | 73.4 | 54.4 | 67.9 | 39.9 | 45.3 | **76.6** |

Table 10: Performance comparison on BEIR English retrieval benchmark.

| Dataset | Size | On MTEBv2 | Large Models (≥1B params) | | | | | | | | Small Models (<1B params) | | | | | | | | |
|---------|------|-----------|----------|----------|----------|----------|-----------|----------|----------|--------|---------|----------|--------------|---------|------------|-----------|---------------|----------|-------|
| | | | Qwen3-8B | bge-en-icl | Qwen3-4B | inf-v1-1.5b | e5-large-inst | gte-large | LENS-d4K | inf-r1 | gte-base | bge-large | gte-base-v1.5 | e5-large | bge-lg-v1.5 | Qwen3-0.6B | opensearch-gte | Splade-V3 | MILCO |
| *Small Collections (<1M documents)* | | | | | | | | | | | | | | | | | | | |
| ArguAna | 8.7K | yes | 76.9 | 83.1 | 75.6 | 81.5 | 58.5 | 57.2 | 77.3 | 84.9 | 57.1 | 62.5 | 63.5 | 54.4 | 64.5 | 71.0 | 52.1 | 50.9 | 61.8 |
| FiQA2018 | 58K | yes | 64.6 | 59.7 | 62.7 | 56.1 | 48.4 | 44.5 | 60.4 | 62.4 | 40.8 | 45.0 | 48.7 | 43.8 | 45.0 | 46.6 | 40.7 | 37.4 | 42.7 |
| NFCorpus | 3.6K | no | 41.5 | 41.9 | 41.1 | 38.6 | 36.3 | 38.2 | 41.6 | 43.7 | 37.9 | 34.6 | 35.9 | 34.0 | 38.1 | 36.7 | 36.0 | 35.7 | 36.3 |
| QuoraRetrieval | 523K | no | 88.9 | 91.0 | 88.1 | 89.6 | 89.2 | 88.3 | 90.8 | 90.4 | 88.2 | 89.0 | 88.4 | 89.3 | 89.1 | 87.8 | 87.3 | 81.4 | 88.2 |
| SCIDOCS | 26K | yes | 32.7 | 25.3 | 31.4 | 26.3 | 19.2 | 23.4 | 27.5 | 30.8 | 23.1 | 22.2 | 21.9 | 17.5 | 22.6 | 24.4 | 16.7 | 15.8 | 16.8 |
| SciFact | 5.2K | no | 78.5 | 79.1 | 78.3 | 82.8 | 71.6 | 74.3 | 78.4 | 85.4 | 76.2 | 72.4 | 76.8 | 70.2 | 74.6 | 69.7 | 72.5 | 71.0 | 70.1 |
| TRECCOVID | 171K | yes | 95.0 | 79.1 | 92.9 | 72.4 | 82.5 | 70.2 | 69.7 | 75.1 | 68.8 | 75.4 | 73.1 | 71.2 | 74.7 | 90.5 | 73.3 | 74.8 | 74.0 |
| Touche2020 | 383K | yes | 35.9 | 30.5 | 35.4 | 21.3 | 27.4 | 25.5 | 25.9 | 24.4 | 22.6 | 26.6 | 25.2 | 23.1 | 24.8 | 33.2 | 39.0 | 29.3 | 28.0 |
| *Large Collections (≥1M documents)* | | | | | | | | | | | | | | | | | | | |
| ClimateFEVER | 5.4M | yes | 47.4 | 45.4 | 47.4 | 41.5 | 29.9 | 28.8 | 44.6 | 41.8 | 28.1 | 38.2 | 40.4 | 25.7 | 36.6 | 42.1 | 31.2 | 32.3 | 30.8 |
| DBPedia | 4.6M | no | 49.7 | 51.6 | 48.2 | 48.6 | 38.4 | 42.4 | 50.1 | 50.4 | 41.2 | 43.9 | 39.9 | 41.3 | 44.1 | 39.5 | 45.5 | 45.0 | 45.1 |
| FEVER | 5.4M | yes | 91.9 | 92.8 | 91.6 | 90.9 | 78.0 | 84.5 | 92.4 | 94.2 | 81.5 | 86.7 | 94.8 | 82.8 | 87.2 | 88.2 | 86.1 | 79.6 | 83.4 |
| HotpotQA | 5.2M | yes | 76.8 | 85.1 | 74.7 | 76.3 | 69.3 | 67.2 | 85.1 | 82.0 | 65.8 | 74.6 | 67.8 | 71.2 | 74.1 | 65.7 | 71.6 | 69.2 | 77.7 |
| MSMARCO | 8.8M | no | 43.6 | 46.8 | 42.7 | 41.0 | 40.4 | 40.9 | 47.0 | 44.1 | 40.2 | 42.6 | 42.6 | 43.7 | 42.5 | 38.0 | 42.6 | 44.0 | 42.0 |
| NQ | 2.7M | no | 65.3 | 73.9 | 63.1 | 64.2 | 57.8 | 54.8 | 73.1 | 69.7 | 52.8 | 53.2 | 53.0 | 64.0 | 55.0 | 53.5 | 58.2 | 58.6 | 64.9 |
| **Avg (All)** | | | 63.5 | 63.2 | 62.4 | 59.4 | 53.3 | 52.9 | 61.7 | 62.8 | 51.7 | 54.8 | 55.1 | 52.3 | 55.2 | 56.2 | 53.8 | 51.1 | 54.4 |
| **Avg (Large)** | | | 62.4 | 66.0 | 61.3 | 60.4 | 52.3 | 53.1 | 65.4 | 63.7 | 51.6 | 56.5 | 56.4 | 54.8 | 56.6 | 54.5 | 55.8 | 53.3 | 57.3 |

## A.3 ENGLISH RETRIEVAL RESULTS (BEIR)

In Table 10, we report the performance of MILCO and baselines on various retrieval tasks evaluated on BEIR English benchmark (Thakur et al., 2021).

On average across BEIR benchmarks, MILCO attains 54.4, slightly behind the dense competitor Qwen3-0.6B (56.2; −1.8). This gap is expected, as Qwen3-0.6B benefits from instruction tuning, which the Qwen3 paper (Zhang et al., 2025) reports adds +1–5%, while MILCO does not use instructions. Compared to other English-only sparse baselines, MILCO is clearly stronger than Splade-V3 (+3.3) and marginally ahead of opensearch-gte (+0.6), while still trailing LENS-d4K (−7.3), a much larger sparse model. It is worth noting, however, that the comparison to Splade-V3 is not entirely fair: Splade is only trained on MSMARCO, while MILCO (and most other baselines) is trained on much larger data, including BEIR's in-domain training sets, which naturally favors transfer to the BEIR evaluation benchmark.

When focusing on the more challenging large-collection datasets (≥1M documents), MILCO shows its main strength. It achieves an average of 57.3, surpassing Qwen3-0.6B (54.5; +2.8), Splade-V3 (53.3; +4.0), and opensearch-gte (55.8; +1.5). MILCO's improvements are particularly pronounced on datasets such as HotpotQA (77.7 vs. 65.7; +12.0) and NQ (64.9 vs. 53.5; +11.4). Although it still falls behind LENS-d4K (65.4; −8.1), the strong performance on large collections is noteworthy because such scenarios are the most relevant to real-world search applications, where corpora often contain millions of documents.

Table 11: Performance of MILCO compared to dense baselines on the LIMIT benchmark.

| Model | Dim | Recall@2 | Recall@10 | Recall@100 |
|---|---|---|---|---|
| BM25 | default | 85.7 | 90.4 | 93.6 |
| GTE-ModernColBERT | default | 23.1 | 34.6 | 54.8 |
| E5-Mistral 7B | 32 | 0 | 0 | 0.5 |
| E5-Mistral 7B | 64 | 0 | 0.1 | 0.4 |
| E5-Mistral 7B | 128 | 0.1 | 0.3 | 1.0 |
| E5-Mistral 7B | 256 | 0.4 | 0.9 | 1.9 |
| E5-Mistral 7B | 512 | 0.7 | 1.3 | 3.8 |
| E5-Mistral 7B | 768 | 0.9 | 1.7 | 4.3 |
| E5-Mistral 7B | 1024 | 0.9 | 1.8 | 5.9 |
| E5-Mistral 7B | 2048 | 1.0 | 1.9 | 6.8 |
| E5-Mistral 7B | 3072 | 1.3 | 2.0 | 7.7 |
| E5-Mistral 7B | 4096 | 1.3 | 2.2 | 8.3 |
| GritLM 7B | 32 | 0 | 0 | 0.8 |
| GritLM 7B | 64 | 0 | 0.1 | 0.3 |
| GritLM 7B | 128 | 0.1 | 0.3 | 1.3 |
| GritLM 7B | 256 | 0.1 | 0.4 | 2.8 |
| GritLM 7B | 512 | 0.6 | 1.8 | 6.5 |
| GritLM 7B | 768 | 1.5 | 3.1 | 8.7 |
| GritLM 7B | 1024 | 1.8 | 3.5 | 10.6 |
| GritLM 7B | 2048 | 2.3 | 4.3 | 11.8 |
| GritLM 7B | 3072 | 2.0 | 4.3 | 12.9 |
| GritLM 7B | 4096 | 2.4 | 4.1 | 12.9 |
| Qwen3-Embed | 32 | 0 | 0.1 | 1.1 |
| Qwen3-Embed | 64 | 0 | 0.2 | 1.0 |
| Qwen3-Embed | 128 | 0.3 | 0.4 | 1.8 |
| Qwen3-Embed | 256 | 0.4 | 0.8 | 3.2 |
| Qwen3-Embed | 512 | 0.6 | 1.3 | 3.3 |
| Qwen3-Embed | 768 | 0.7 | 1.5 | 3.8 |
| Qwen3-Embed | 1024 | 0.7 | 1.6 | 4.6 |
| Qwen3-Embed | 2048 | 0.9 | 1.7 | 4.7 |
| Qwen3-Embed | 3072 | 0.8 | 1.6 | 4.8 |
| Qwen3-Embed | 4096 | 0.8 | 1.8 | 4.8 |
| Gemini-Embed | 2 | 0 | 0 | 0.1 |
| Gemini-Embed | 4 | 0 | 0 | 0.0 |
| Gemini-Embed | 8 | 0 | 0 | 0.0 |
| Gemini-Embed | 16 | 0 | 0 | 0.0 |
| Gemini-Embed | 32 | 0 | 0 | 0.0 |
| Gemini-Embed | 64 | 0 | 0 | 0.3 |
| Gemini-Embed | 128 | 0 | 0.1 | 0.3 |
| Gemini-Embed | 256 | 0 | 0.1 | 1.2 |
| Gemini-Embed | 512 | 0.2 | 1.1 | 3.6 |
| Gemini-Embed | 768 | 0.9 | 2.5 | 7.6 |
| Gemini-Embed | 1024 | 1.3 | 2.7 | 8.1 |
| Gemini-Embed | 2048 | 1.5 | 3.1 | 8.5 |
| Gemini-Embed | 3072 | 1.6 | 3.5 | 10.0 |
| ① **MILCO** (SAP, $SCT_{KD}$, LexEcho) | 280,524 | 26.2 | 47.0 | 73.5 |

## A.4 EMBEDDING LIMIT TEST

Table 11 shows the advantage of MILCO over strong state-of-the-art dense baselines on the LIMIT test (Weller et al., 2025). MILCO achieves an R@100 of 73.5, whereas dense models such as Gemini-Embed and Qwen3-Embed nearly collapse to zero.

## A.5 CORRELATION BETWEEN TEXT LENGTH AND VECTOR SPARSITY

In Figure 7, we show a strong correlation between input length and the sparsity of vectors produced by MILCO. Unlike dense retrieval methods, which always generate fixed-length vectors for all queries and documents, LSR methods, including MILCO, adaptively determine the optimal sparsity

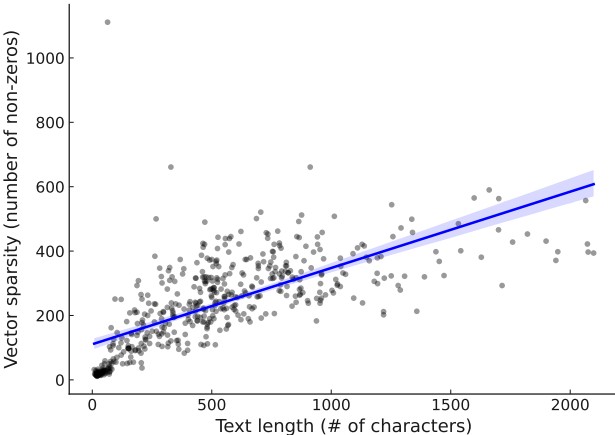

Figure 7: MILCO: Correlation between input text length and the sparsity of output vectors.

Table 12: MILCO: Effectiveness at different TopK (tokens). (nDCG@10, MIRACL)

| TopK (tokens) | Avg | ar | bn | de | en | es | fa | fi | fr | hi | id | ja | ko | ru | sw | te | th | yo | zh |
|---|---|---|---|---|---|---|---|---|---|---|---|---|---|---|---|---|---|---|---|
| 10 | 44.5 | 55.66 | 50.1 | 38.38 | 34.15 | 34.6 | 35.65 | 57.53 | 40.42 | 29.65 | 38.16 | 43.33 | 49.13 | 42.65 | 55.92 | 56.41 | 48.56 | 53.11 | 37.9 |
| 20 | 54.6 | 66.81 | 62.26 | 45.75 | 43.32 | 41.98 | 45.07 | 66.9 | 47.02 | 41.09 | 45.72 | 55.88 | 57.58 | 52.69 | 65.76 | 72.89 | 64.93 | 60.99 | 45.97 |
| 50 | 63.8 | 74.29 | 73.7 | 52.87 | 51.41 | 51.65 | 52.95 | 74.29 | 53.31 | 52.79 | 53.08 | 67.11 | 66.08 | 63.95 | 74.1 | 82.61 | 77.64 | 69.69 | 56.82 |
| 100 | 68.4 | 77.52 | 77.98 | 58.07 | 56.8 | 56.36 | 59.31 | 77.2 | 58.49 | 60.09 | 57.27 | 72.11 | 69.02 | 69.34 | 76.93 | 85.13 | 81.45 | 76.12 | 61.79 |
| 200 | 71.0 | 79.63 | 80.59 | 60.25 | 59.7 | 59.87 | 61.71 | 79.7 | 61.41 | 63.8 | 59.46 | 75.59 | 70.7 | 72.76 | 78.72 | 87.05 | 83.15 | 79.75 | 64.72 |
| 300 | 72.1 | 80.06 | 81.46 | 61.49 | 61.04 | 61.05 | 62.75 | 80.27 | 62.55 | 66.11 | 60.33 | 76.43 | 71.3 | 73.59 | 79.72 | 87.45 | 83.93 | 82.36 | 66.14 |
| 500 | 72.6 | 80.58 | 81.65 | 62.39 | 61.98 | 61.86 | 63.45 | 80.68 | 62.75 | 66.05 | 61.06 | 76.94 | 72.12 | 74.45 | 80.02 | 87.92 | 84.34 | 82.26 | 66.85 |
| 700 | 72.7 | 80.83 | 81.97 | 62.75 | 62.1 | 62.14 | 63.13 | 80.7 | 62.79 | 65.21 | 60.97 | 77.28 | 72.03 | 74.8 | 80.16 | 87.95 | 84.41 | 82.19 | 66.99 |
| 1000 | 72.7 | 80.81 | 82.12 | 62.83 | 62.09 | 62.17 | 63.04 | 80.63 | 62.74 | 65.12 | 60.99 | 77.28 | 71.94 | 74.89 | 80.26 | 87.75 | 84.48 | 82.36 | 66.94 |

Table 13: MILCO: Effectiveness at different pruning percentile $P$. (nDCG@10, MIRACL)

| P | #Tokens | Avg | ar | bn | de | en | es | fa | fi | fr | hi | id | ja | ko | ru | sw | te | th | yo | zh |
|---|---|---|---|---|---|---|---|---|---|---|---|---|---|---|---|---|---|---|---|---|
| 10 | 512.5 | 72.6 | 80.8 | 82.0 | 62.7 | 62.1 | 62.0 | 63.1 | 80.6 | 62.8 | 65.1 | 60.9 | 77.3 | 72.0 | 74.9 | 80.2 | 87.8 | 84.4 | 82.0 | 66.8 |
| 20 | 455.8 | 72.6 | 80.7 | 81.8 | 62.8 | 62.0 | 61.8 | 63.2 | 80.6 | 62.9 | 64.8 | 61.0 | 77.3 | 72.0 | 74.9 | 80.2 | 87.8 | 84.5 | 81.7 | 66.8 |
| 50 | 285.6 | 72.3 | 80.7 | 81.8 | 61.7 | 61.1 | 61.7 | 63.1 | 80.5 | 62.6 | 64.3 | 60.8 | 76.9 | 71.9 | 74.6 | 80.1 | 87.8 | 84.2 | 80.4 | 66.4 |
| 70 | 172.2 | 71.7 | 80.3 | 81.5 | 60.9 | 60.1 | 60.5 | 62.5 | 79.9 | 61.3 | 64.2 | 60.1 | 76.6 | 70.5 | 74.2 | 79.4 | 87.8 | 83.8 | 80.7 | 65.2 |
| 80 | 115.1 | 70.6 | 79.6 | 80.7 | 60.9 | 59.0 | 59.6 | 61.3 | 78.8 | 60.8 | 62.3 | 58.7 | 75.4 | 68.8 | 73.0 | 78.3 | 86.9 | 83.7 | 79.2 | 64.0 |
| 85 | 86.4 | 69.5 | 78.6 | 79.5 | 59.0 | 57.2 | 58.5 | 60.1 | 78.3 | 59.0 | 61.7 | 58.4 | 74.2 | 67.5 | 71.7 | 77.2 | 86.4 | 82.8 | 77.7 | 63.0 |
| 90 | 57.9 | 67.4 | 76.6 | 77.8 | 57.8 | 55.0 | 56.1 | 56.1 | 76.7 | 57.4 | 58.6 | 56.4 | 72.1 | 65.7 | 68.9 | 76.3 | 84.5 | 81.4 | 75.0 | 60.7 |
| 95 | 29.2 | 62.2 | 71.8 | 71.8 | 52.5 | 48.7 | 51.2 | 48.9 | 71.8 | 53.6 | 52.6 | 52.5 | 65.5 | 63.3 | 63.3 | 71.3 | 78.1 | 76.2 | 70.5 | 55.7 |
| 97 | 17.7 | 56.0 | 66.2 | 65.9 | 44.2 | 42.4 | 46.6 | 44.2 | 66.0 | 49.4 | 45.0 | 48.5 | 57.4 | 58.3 | 56.1 | 65.0 | 67.4 | 68.6 | 67.3 | 50.0 |
| 99 | 6.5 | 39.5 | 48.38 | 46.14 | 32.18 | 29.04 | 33.31 | 28.55 | 50.53 | 37.56 | 28.06 | 37.17 | 38.38 | 42.27 | 37.82 | 43.04 | 43.07 | 44.59 | 57.09 | 34.13 |

(i.e., the number of non-zero elements) based on the content density of the input text, as approximated by its length. This allows MILCO to allocate fewer tokens for short texts and more for longer ones, while maintaining the same average sparsity overall.

On Table 12 and Table 13, we show the results of two different post-hoc pruning methods (Top-k Pruning and Mass-based Pruning) applied on MILCO.

## A.6 ALIGNMENT AND CONTRASTIVE TRAINING DATA

The sources and statistics of the data used for our Sparse Alignment Pretraining are reported in Table 14. In total, the corpus consists of 594 million bi-text pairs, each containing one English sentence and one non-English sentence with the same semantic meaning.

The statistics of the data used for our contrastive training are shown in Table 15. This dataset contains 1.4M queries collected from 16 datasets, covering English, Chinese, and 16 additional languages from Mr.TYDI (Zhang et al., 2021) and MIRACL (Zhang et al., b). Similar to prior work (Chen et al., 2024; Li et al.; Lei et al., 2025), our training data also includes many in-domain datasets from BEIR (Thakur et al., 2021).

Table 14: Pretraining datasets: Parallel Sentences collected from OPUS by Reimers & Gurevych (2019).

| Dataset Name | #Pairs |
|---|---|
| mmarco (passages) | 115M |
| wikititles | 14M |
| wikimatrix | 19M |
| europarl | 50M |
| opensubtitles | 274M |
| talks | 20M |
| tatoeba | 8M |
| jw300 | 92M |
| news-commentary | 2M |
| **Total** | **594M** |

Table 15: Contrastive Training Data obtained from Chen et al. (2024).

| Dataset | #Samples |
|---|---|
| en_msmarco (Nguyen et al., 2016) | 485,823 |
| en_eli5 (Fan et al., 2019) | 150,000 |
| zh_mmarco_zh (Bonifacio et al., 2021) | 100,000 |
| zh_t2ranking (Xie et al., 2023) | 90,467 |
| en_squad (Rajpurkar et al., 2016) | 87,599 |
| en_hotpotqa (Yang et al., 2018) | 84,516 |
| zh_dureader (He et al., 2017) | 80,416 |
| en_trivia (Joshi et al., 2017) | 60,315 |
| en_quora (Sharma et al., 2019) | 60,202 |
| en_nq (Kwiatkowski et al., 2019) | 58,568 |
| multilingual_mrtydi (Zhang et al., 2021) | 48,729 |
| multilingual_miracl (Zhang et al., b) | 40,203 |
| en_fever (Thorne et al., 2018) | 29,096 |
| en_fiqa (Maia et al., 2018) | 5,500 |
| en_arguana (Wachsmuth et al., 2018) | 4,065 |
| en_scidocs (Cohan et al., 2020) | 884 |
| **Total** | **1,386,383** |

## A.7 CONTRASTIVE TRAINING: DISTILLATION

For each query $q_i$, we consider a document candidate set consisting of one positive $d^+$ and a set of negatives $D^-$. The precomputed teacher scores are from the cross-encoder, denoted as $\theta_{\text{CE}}(d, q)$. The student scores $\theta_{\text{milco}}(d, q)$ are estimated via the dot product of MILCO's lexical representations. These scores are converted into distributions over the candidate set with a softmax:

$$P_x(d|q) = \frac{\exp\big(\theta_x(d, q)\big)}{\sum_{d' \in \{d^+, D^-\}} \exp\big(\theta_x(d', q)\big)}, \tag{9}$$

where $\theta_x(q, d)$ denotes either the teacher or student scoring function of the given query and document. The distillation objective is then defined as the KL divergence between the teacher's and the student's distributions across a batch of $B$ queries:

$$L_{\text{KLD}} = \frac{1}{B} \sum_{i=1}^{B} \mathbf{KL}\big(P_{\text{milco}}(d|q_i) || P_{\text{CE}}(d|q_i)\big). \tag{10}$$

A.8  TRAINING CONFIGURATIONS AND HYPER-PARAMETERS

The hyperparameters for pretraining and training are reported in Table 16 and Table 17. Both training stages are conducted on 16 GPU nodes, each equipped with 8 AMD Instinct MI250X GPU dies. We instantiate the Multilingual Connector with a simple randomly-initialized MLP layer with a GELU activation function. For sparse regularization, we set the regularization weight to $1e{-}5$ for both queries and documents during contrastive training. We train MILCO using the HuggingFace framework (Wolf et al., 2019). Hyperparameters not listed in Table 16 and Table 17 are set to the default values defined in HuggingFace's `TrainingArguments`.

Table 16: MILCO: Hyperparameters for Sparse Alignment Pre-training.

| Hyperparameter | Value |
| --- | --- |
| training_type | alignment |
| model_type | bert |
| lsr_encoder_checkpoint | naver/splade-v3 |
| multilingual_encoder_checkpoint | BAAI/bge-m3-unsupervised |
| train_datasets | mmarco, wikititles, wikimatrix, europarl, opensubtitles, talks, tatoeba, jw300, news-commentary |
| seed | 42 |
| max_length | 256 |
| per_device_train_batch_size | 64 |
| per_device_eval_batch_size | 128 |
| num_train_epochs | 2 |
| save_total_limit | 2 |
| warmup_steps | 10000 |
| lr_scheduler_type | cosine |
| dataloader_num_workers | 8 |
| learning_rate | 2e-5 |
| bf16 | True |
| logging_steps | 500 |
| save_steps | 20000 |
| pooling | max |
| remove_unused_columns | False |
| dynamic_length | True |

Table 17: MILCO: Hyperparameters for Sparse Contrastive Training.

| Hyperparameter | Value |
| --- | --- |
| training_type | distillation |
| model_type | bert |
| lsr_encoder_checkpoint | naver/splade-v3 |
| multilingual_encoder_checkpoint | BAAI/bge-m3-unsupervised |
| train_group_size | 8 |
| lambda_q | 1e-3 |
| lambda_d | 1e-5 |
| train_datasets | bge |
| seed | 42 |
| max_length | 512 |
| per_device_train_batch_size | 8 |
| per_device_eval_batch_size | 32 |
| num_train_epochs | 8 |
| save_total_limit | 2 |
| warmup_ratio | 0.03 |
| lr_scheduler_type | cosine |
| dataloader_num_workers | 1 |
| learning_rate | 2e-5 |
| bf16 | True |
| logging_steps | 500 |

## A.9    LIST OF LANGUAGES SUPPORTED BY MILCO

Table 18: Datasets and their supported languages.

| Dataset | Languages (standardized) | #languages |
|---------|--------------------------|------------|
| MIRACL | ar, bn, de, en, es, fa, fi, fr, hi, id, ja, ko, ru, sw, te, th, yo, zh | 18 |
| MLDR | ar, de, en, es, fr, hi, it, ja, ko, pt, ru, th, zh | 13 |
| MKQA | ar, da, de, es, fi, fr, he, hu, it, ja, km, ko, ms, nl, no, pl, pt, ru, sv, th, tr, vi, zh-cn, zh-hk, zh-tw | 25 |
| BelebeleRetrieval | acm, af, en | 3 |
| MLQARetrieval | ar, de, en, es, hi, vi | 6 |
| TwitterHjerneRetrieval | dan | 1 |
| WikipediaRetrievalMultilingual | bg, bn, cs, da, de, en, fa, fi, hi, it, nl, no, pt, ro, sr, sv | 16 |
| WikiMatrix | ar, bg, ca, cs, da, de, el, es, et, fa, fi, fr, gl, he, hi, hr, hu, hy, id, it, ja, ka, ko, lt, lv, mk, ms, nl, pl, pt, ro, ru, sk, sl, sq, sr, sv, th, tr, uk, ur, vi, zh-cn | 41 |
| parallel-sentences-opensubtitles | ar, bg, ca, cs, da, de, el, es, et, fa, fi, fr, gl, he, hi, hr, hu, id, it, ja, ka, ko, lt, mk, mr, nl, pl, pt, ro, ru, sk, sl, sq, sr, sv, tr, uk, vi, zh | 38 |
| parallel-sentences-tatoeba | ar, bg, ca, cs, da, de, el, es, et, fa, fi, fr, gl, gu, he, hi, hr, hu, hy, id, it, ja, ka, ko, ku, lt, lv, mk, mn, mr, ms, my, nb, nl, pl, pt, ro, ru, sk, sl, sq, sr, sv, th, tr, uk, ur, vi, zh | 46 |
| parallel-sentences-global-voices | ar, bg, ca, cs, da, de, el, es, fa, fr, he, hi, hu, id, it, ko, mk, my, nl, pl, pt, ro, ru, sq, sr, sv, tr, ur | 27 |
| parallel-sentences-europarl | bg, cs, da, de, el, es, et, fi, fr, hu, it, lt, lv, nl, pl, pt, ro, sk, sl, sv | 20 |
| parallel-sentences-talks | ar, bg, ca, cs, da, de, el, es, et, fa, fi, fr, fr-ca, gl, gu, he, hi, hr, hu, hy, id, it, ja, ka, ko, ku, lt, lv, mk, mn, mr, ms, my, nb, nl, pl, pt, pt-br, ro, ru, sk, sl, sq, sr, sv, th, tr, uk, ur, vi, zh-cn, zh-tw | 50 |
| parallel-sentences-jw300 | ar, bg, cs, da, de, el, es, et, fa, fi, fr, gu, he, hi, hr, hu, hy, id, it, ja, ka, ko, lt, lv, mk, mn, mr, my, nl, pl, pt, ro, ru, sk, sl, sq, sr, sv, th, tr, uk, ur, vi | 43 |
| parallel-sentences-news-commentary | ar, cs, de, es, fr, it, ja, nl, pt, ru | 10 |
| mmarco | ar, zh, nl, en, fr, de, hi, id, it, ja, pt, ru, es, vi | 14 |
| All Test Datasets | acm, af, ar, bg, bn, cs, da, de, en, es, fa, fi, fr, he, hi, hu, id, it, ja, km, ko, ms, nl, no, pl, pt, ro, ru, sr, sv, sw, te, th, tr, vi, yo, zh, zh-cn, zh-hk, zh-tw | 39 |
| All (Pretrain/Train + Test) datasets | acm, af, ar, bg, bn, ca, cs, da, de, el, en, es, et, fa, fi, fr, fr-ca, gl, gu, he, hi, hr, hu, hy, id, it, ja, ka, km, ko, ku, lt, lv, mk, mn, mr, ms, my, nb, nl, no, pl, pt, pt-br, ro, ru, sk, sl, sq, sr, sv, sw, te, th, tr, uk, ur, vi, yo, zh, zh-cn, zh-hk, zh-tw | 63 |

## A.10    ABLATIONS ON LEXECHO HEAD.

To provide additional insights on our LexEcho head, we perform an ablation that interpolates between the pivot (English) and source views in LexEcho. Specifically, we multiply the English view by $\alpha$ and the source view by $(1 - \alpha)$, with $\alpha \in \{0.0, 0.2, \dots, 1.0\}$, and report MIRACL results in Table 19.

Table 19: MILCO performance under different pivot–source weighting schemes on the MIRACL (hard negatives) benchmark.

| $\alpha$ | ar | bn | en | es | fa | fi | fr | hi | id | ja | ko | ru | sw | te | th | zh | de | yo | Avg |
|---|---|---|---|---|---|---|---|---|---|---|---|---|---|---|---|---|---|---|---|
| 0.00 | 5.21 | 9.61 | 3.56 | 0.09 | 1.74 | 8.18 | 0.14 | 1.56 | 4.71 | 3.33 | 11.14 | 2.19 | 2.75 | 31.35 | 14.90 | 0.00 | 0.20 | 0.27 | 5.61 |
| 0.20 | 11.01 | 17.51 | 6.16 | 0.91 | 4.23 | 17.80 | 1.51 | 3.95 | 9.55 | 7.16 | 19.09 | 5.35 | 6.95 | 43.42 | 22.47 | 0.11 | 1.84 | 4.49 | 10.19 |
| 0.40 | 79.38 | 79.30 | 56.55 | 55.89 | 57.29 | 79.72 | 60.70 | 57.46 | 58.46 | 75.36 | 71.99 | 73.10 | 77.52 | 86.77 | 82.44 | 56.95 | 59.67 | 63.23 | 68.43 |
| 0.50 | 80.79 | 82.00 | 62.08 | 62.06 | 63.08 | 80.64 | 62.73 | 65.06 | 60.91 | 77.25 | 72.00 | 74.96 | 80.18 | 87.82 | 84.46 | 66.80 | 62.83 | 82.14 | 72.66 |
| 0.60 | 80.08 | 81.43 | 62.03 | 62.00 | 62.34 | 79.91 | 63.14 | 64.94 | 60.14 | 76.64 | 71.13 | 74.30 | 79.61 | 87.11 | 83.81 | 65.54 | 61.73 | 81.72 | 72.09 |
| 0.80 | 78.19 | 80.22 | 60.18 | 61.27 | 60.36 | 78.81 | 62.16 | 64.00 | 58.39 | 74.66 | 69.12 | 72.95 | 78.49 | 84.05 | 81.47 | 63.63 | 60.58 | 80.86 | 70.52 |
| 1.00 | 77.77 | 79.35 | 59.53 | 60.98 | 59.73 | 78.43 | 62.19 | 63.31 | 57.97 | 73.80 | 68.52 | 72.74 | 78.04 | 82.90 | 80.72 | 63.33 | 60.55 | 80.46 | 70.02 |

We find that MILCO performs very poorly with only the source view ($\alpha = 0.0$; average nDCG@10 $= 5.61$), indicating that the English pivot is essential. With only the English view ($\alpha = 1.0$), MILCO is already strong (average nDCG@10 $= 70.02$) but not optimal. The best result is obtained with a roughly balanced fusion ($\alpha = 0.5$), reaching an average nDCG@10 of 72.66 (around 3–4% relative improvement over $\alpha = 1.0$). Performance for $\alpha$ between 0.5 and 0.6 is very similar, suggesting that LexEcho is not overly sensitive to the exact weighting as long as both views contribute.

## A.11 Effect of Language Coverage in Alignment Pretraining

We now investigate how language coverage in the parallel corpus used for MILCO's sparse alignment pretraining affects the final retrieval performance across MIRACL languages. To this end, we compare MILCO (alignment + distillation) against our best-performing dense baseline (distillation only), trained with the same data and compute, and report per-language $\Delta$nDCG@10. We then aggregate languages by their share of alignment data to analyze the relationship between coverage and effectiveness (Tables 20 and 21).

Table 20: Average $\Delta$nDCG@10 as a function of alignment coverage. Languages are grouped into well-represented ($P \geq 1\%$) and under-represented ($P < 1\%$) buckets based on their proportion in the parallel corpus.

| Percentage category: | $P \geq 1$ | $P < 1$ |
|---|---|---|
| Avg. $\Delta$nDCG@10 | 1.59 | 0.77 |

Table 21 lists nDCG@10 and $\Delta$nDCG@10 for all 18 MIRACL languages, together with their proportion in the parallel corpus. Aggregating by coverage (Table 20), we observe an average gain of $+1.59$ $\Delta$nDCG@10 for well-represented languages ($P \geq 1\%$) and a smaller but still positive average gain of $+0.77$ for under-represented languages ($P < 1\%$), including languages with 0 parallel samples (sw, te, yo, bn). The main exception is fa ($-1.09$), which we plan to analyze further.

These results indicate that higher coverage in the alignment corpus amplifies the gains from MILCO, but is not strictly required: even languages that are weakly covered or entirely absent from the alignment corpus still benefit on average. This behavior highlights MILCO's multilingual alignment pretraining as an effective mechanism for knowledge transfer and sharing across languages.

Table 21: MILCO effectiveness vs. alignment data distribution across languages (MIRACL Hard Negatives.). We report $\mathrm{nDCG@10}$, per-language $\Delta\mathrm{nDCG@10}$ (MILCO vs. dense baseline), and the number and proportion of parallel corpus samples used for alignment pretraining.

| Lang | nDCG@10 | $\Delta$nDCG@10 | Samples | Percentage |
|---|---|---|---|---|
| ar | 80.4 | 0.919 | 19002229 | 5.5 |
| bn | 82.6 | 2.387 | 0 | 0 |
| en | 60.4 | 3.585 | – | – |
| es | 60.9 | 0.487 | 28470451 | 8.23 |
| fa | 62.3 | -1.09 | 646913 | 0.19 |
| fi | 81.2 | 2.676 | 4958793 | 1.43 |
| fr | 61.7 | -0.922 | 22574836 | 6.53 |
| hi | 64.4 | 2.158 | 9626940 | 2.78 |
| id | 60.9 | 2.498 | 17591514 | 5.09 |
| ja | 77.2 | 2.524 | 11542221 | 3.34 |
| ko | 72.1 | 1.548 | 3648265 | 1.05 |
| ru | 74.6 | 2.458 | 16592711 | 4.8 |
| sw | 80.3 | 0.734 | 0 | 0 |
| te | 87.9 | 0.87 | 0 | 0 |
| th | 84.2 | 1.295 | 1668858 | 0.48 |
| zh | 65.5 | 1.328 | 9006690 | 2.6 |
| de | 61.4 | 1.823 | 24431450 | 7.07 |
| yo | 83.6 | 0.41 | 0 | 0 |

