# OpenReview forum: "MILCO: Learned Sparse Retrieval Across Languages via a Multilingual Connector"
_ICLR.cc/2026/Conference — ICLR 2026 Poster_

### Official Review · Reviewer_4uhs · 2025-10-31

**Soundness:** 3
**Presentation:** 2
**Contribution:** 2
**Rating:** 4
**Confidence:** 4

**Summary:**

This paper proposes MILCO, a multilingual learned-sparse retriever that projects inputs from many languages into a shared English lexical space via a lightweight multilingual connector. The model is trained in two stages: Sparse Alignment Pretraining (SAP) using large bitext corpora and an English LSR teacher, followed by Sparse Contrastive Training (SCT) with KL-distillation and InfoNCE. To preserve rare source-language entities lost in projection, the authors propose LexEcho, a dual-view head combining an English view with a weighted source view via an ECHO token. The method is empirically evaluated on MIRACL, MTEBv2, MLDR, MKQA and shows strong performance, supporting LSR post-hoc pruning and interpretability.

**Strengths:**

1. **Addresses an important practical challenge**: multilingual sparse retrieval with interpretability.
2. **Good technical contributions**: SAP (sparse-aware MSE on logits) and LexEcho (dual view + [ECHO]) address semantic collapse of bi-text logits mapping.
3. **Strong empirical results** across multiple benchmarks, with efficiency/accuracy tradeoff analysis (post-hoc pruning).

**Weaknesses:**

1. **Motivation needs to be strengthened**: why choosing to map multilingual tokens to English tokens? Could the authors clarify the rationale for specifically choosing English as the shared lexical space? What makes the English vocabulary a preferable or necessary choice in this framework?
2. **Necessity outside of cross-lingual bi-text retrieval**: Does the proposed alignment (via the English teacher model) offer empirical or conceptual benefits that would not hold if the shared space were defined in another language or script? For purely monolingual retrieval scenarios (e.g., Chinese-to-Chinese), is the English-space projection still beneficial, or does it primarily target cross-lingual retrieval only?
3. **Experimental comparison could be more rigorous**: disclosure of training compute, teacher model details, and dataset overlaps with baselines is incomplete. Without that, claims of beating much larger models risk being influenced by differing training data/compute.
4. **Robustness to low-resource languages and domain shifts is not fully analyzed**: SAP relies on parallel corpora which may bias alignment.
5. **Engineering/operational metrics (index size, latency, end-to-end retrieval QPS)** are not fully reported but are crucial for claims about effectiveness and deployability.

**Questions:**

1. Please answer **Weakness 1** by clarifying the motivation for mapping multilingual tokens to English tokens.
2. Please answer **Weakness 2** by clarifying the necessity of your design except for cross-lingual bi-text retrieval. Any related comparison results or experiments are welcome.
3. Please provide a more detailed training budget (GPU type/count, total GPU hours) and distillation/teacher details (which teacher checkpoints, how produced).
4. How does MILCO perform on truly low-resource languages not well represented in parallel corpus? Can the authors report per-language numbers for the lowest-performing languages?
5. What are the index size and query latency numbers (both full representation and after pruning) using typical inverted index settings (e.g. Anserini with Lucene9) and hardware?
6. Can you provide more ablations on LexEcho fusion (e.g., varying weight between English and source views) and present error types where LexEcho hurts ranking?

---

> ### Author Response · Authors · 2025-11-21
> **Response to Reviewer 4uhs (1/6)**
>
> > why choosing to map multilingual tokens to English tokens? Could the authors clarify the rationale for specifically choosing English as the shared lexical space? What makes the English vocabulary a preferable or necessary choice in this framework?
>
> Our framework does not rely on any special property of English: in principle, MILCO is pivot-language agnostic and could project into any shared lexical space given sufficient alignment data. We chose English for practical reasons: (i) it is the only language for which large-scale, high-quality parallel data between a pivot language and other languages is widely available [1,2], enabling training of the Multilingual Connector, and (ii) most existing LSR methods are developed in English, and our alignment stage requires a strong LSR model to use as an alignment teacher, which makes the English lexical space a natural choice.  In addition, there is evidence that multilingual LLMs operate in a “concept space” that lies closer to English than to other languages [3, 4]. Crucially, MILCO is not forced to rely solely on the pivot language: via the LexEcho head, it always maintains an original-language view alongside the projected pivot-language view, and our experiments show that this dual-view design improves robustness and reduces over-reliance on the pivot-language lexical space.
>
> [1] Schwenk et al., ACL 2021. CCMatrix: Mining Billions of High-Quality Parallel Sentences on the Web
>
> [2] Schwenk et al., EACL 2021. WikiMatrix: Mining 135M Parallel Sentences in 1620 Language Pairs from Wikipedia
>
> [3] Chris et al., ACL 2024.  Do Llamas Work in English? On the Latent Language of Multilingual Transformers
>
> [4] Zhaofeng et al., ICLR 2025.  The Semantic Hub Hypothesis: Language Models Share Semantic Representations Across Languages and Modalities

---

> ### Author Response · Authors · 2025-11-21
> **Response to Reviewer 4uhs (2/6)**
>
> > Does the proposed alignment (via the English teacher model) offer empirical or conceptual benefits that would not hold if the shared space were defined in another language or script? For purely monolingual retrieval scenarios (e.g., Chinese-to-Chinese), is the English-space projection still beneficial, or does it primarily target cross-lingual retrieval only?
>
> For purely monolingual retrieval in **a single language** with abundant in-language supervision (e.g., Chinese–Chinese only), a dedicated monolingual LSR model might indeed be a reasonable choice, as you suggest, since it does not need to share capacity across many languages. Our goal, however, is different: MILCO is designed as a universal model that can handle both monolingual and cross-lingual retrieval across many languages with a single set of parameters, leveraging multilingual training data for knowledge share/transfer. Empirically, MILCO achieves strong monolingual performance (including on Chinese), outperforming dense baselines that perform latent space alignment while simultaneously supporting cross-lingual retrieval. This indicates that projecting into an English lexical space does not preclude learning effective in-language sparse representations.
>
> This approach of projecting into a shared lexical space is central to MILCO’s design, but it is not specific to English. We chose an English teacher because it is a high-resource language with well-established sparse lexical retrieval models and large-scale relevance data, which makes it a strong and convenient pivot. Conceptually, the same alignment mechanism could use any language with a strong teacher and sufficient parallel data; our method does not rely on any linguistic property unique to English.

---

> ### Author Response · Authors · 2025-11-21
> **Response to Reviewer 4uhs (3/6)**
>
> > Disclosure of training compute, teacher model details, and dataset overlaps with baselines is incomplete. Without that, claims of beating much larger models risk being influenced by differing training data/compute.
>
> The full details of the training data used for both alignment and distillation are provided in Tables 13 and 14 in Appendix A.6. Briefly, the alignment data consists of 594 million public parallel translation pairs mapping English sentences to 50+ languages. This alignment corpus draws from multiple sources, including Wikipedia, movie subtitles, news, and translated MS MARCO passages.
>
> The distillation dataset contains ~1.4 million queries, each accompanied by positive and negative documents released by the FlagEmbedding team that produced the BGE model. It covers English data (e.g., MS MARCO, BEIR training splits), Chinese data (e.g., mMARCO, DuReader, T2Ranking), and monolingual retrieval datasets from 18 languages included in the MIRACL and Mr. TyDi training data. As described in the main text, we use SPLADE-v3 as the English LSR teacher for alignment training and bge-reranker-v2.5-gemma2-lightweight as the teacher for distillation.
>
> The complete set of hyperparameters and hardware specifications (including GPU type and count) is reported in Appendix A.8. Alignment pretraining was conducted on 16 GPU nodes, each with 8 AMD MI250x GPU chips, for 36 hours. Distillation was performed on the same cluster and took around 5 hours. We use our proposed SparseMSE loss for the sparse alignment step, and the standard KL loss for distillation together with an L1 loss for sparsity regularization.
>
> While MILCO outperforms models like Qwen3 8B in some experiments, the Qwen3 embedding paper does not provide enough information to directly compare the two in terms of training data and compute [1]. To provide an additional comparison point, we trained a dense baseline using the same training data and compute as MILCO as shown in Table 1. We observe that MILCO achieves a higher average nDCG@10 on the MIRACL benchmark.
>
> **Table 1. Comparison between MILCO and dense baselines under the same training setup (Alignment + Distillation), evaluated on MIRACL benchmark (hard negatives).**
> | Method                                | Size | ar   | bn   | en   | es   | fa   | fi   | fr   | hi   | id   | ja   | ko   | ru   | sw   | te   | th   | zh   | de   | yo   | avg |
> |---------------------------------------|------|------|------|------|------|------|------|------|------|------|------|------|------|------|------|------|------|------|------|------|
> | Dense (Alignment + KL Distillation)   | 561M | 77.0 | 76.6 | 55.3 | 57.5 | 60.2 | 77.1 | 59.0 | 60.5 | 55.5 | 70.5 | 70.9 | 67.5 | 74.4 | 86.1 | 80.6 | 62.5 | 57.6 | 72.8 | 67.9 |
> | Dense (KL Distillation)               | 561M | 79.5 | 80.2 | 56.8 | 60.4 | 63.4 | 78.5 | 62.6 | 62.2 | 58.4 | 74.7 | 70.5 | 72.1 | 79.6 | 87.0 | 82.9 | 64.2 | 59.5 | 83.2 | 70.9 |
> | MILCO (Alignment + InfoNCE)           | 561M | 79.7 | 81.1 | 57.5 | 57.5 | 60.8 | 80.4 | 58.0 | 63.2 | 58.7 | 75.4 | 71.3 | 72.8 | 77.5 | 87.1 | 83.0 | 61.0 | 59.5 | 81.8 | 70.3 |
> | MILCO (Alignment + KL Distillation)   | 561M | 80.8 | 82.0 | 62.1 | 62.1 | 63.1 | 80.6 | 62.7 | 65.1 | 60.9 | 77.3 | 72.0 | 75.0 | 80.2 | 87.8 | 84.5 | 66.8 | 62.8 | 82.1 | 72.7 |
>
> [1] Zhang et al., arXiv 2025. Qwen3 Embedding: Advancing Text Embedding and Reranking Through Foundation Models.

---

> ### Author Response · Authors · 2025-11-21
> **Response to Reviewer 4uhs (4/6)**
>
> > What are the index size and query latency numbers (both full representation and after pruning) using typical inverted index settings (e.g. Anserini with Lucene9) and hardware?
>
> We report index statistics and retrieval latency with MILCO under an inverted-index setting. We use Seismic [1], an ANN method built on top of an inverted index (conceptually similar to Lucene/Anserini, but using approximate search to reduce latency analogously to Faiss). All results are on MIRACL (English subset, about 32M passages), with retrieval run on a single AMD EPYC 7763 CPU core. We build several indexes of the pivot view using different amounts of pruning. The unpruned index (p = 0) has an average posting-list length of 4636.09, resulting in a 61 GB index and an average retrieval latency of 1.85 ms/query. We then prune the lowest-weight dimensions whose cumulative weights account for $p \in \[ 10, 30, 50, 60 \]% $ of the total, which shrinks the inverted index and speeds up retrieval. Results are shown in Table 2.
>
> Regarding index size, post-hoc pruning substantially shrinks the index: 40 GB at $p = 10$, 25 GB at $p = 30$, 16 GB at $p = 50$, and 12 GB at $p = 60$. Thus, at the most aggressive pruning level, we reduce index size by roughly 80% while keeping strong retrieval effectiveness. To contextualize these index-sizes, we compare against Qwen3-Embedding-0.6B with a Faiss dense HNSW index [2] (M = 32, ef = 64) on the same hardware and collection. This dense baseline has an index size of 134 GB, whereas MILCO’s pruned Seismic index is already smaller at $p = 10$ (40 GB vs. 134 GB) and becomes 5–10× smaller at higher pruning levels (25 GB at $p = 30$, 12 GB at $p = 60$).
>
> Regarding latency, pruning also yields consistent improvements over the full representation: $p = 10$ reduces average latency from 1.85 ms to 1.29 ms (~30% speed-up) with virtually no loss in nDCG@10 (56.4 → 56.3), $p = 30$ makes queries about 3× faster (0.61 ms, nDCG@10 = 54.4), and even $p = 60$ achieves a >4× speed-up (0.44 ms) while remaining competitive (nDCG@10 = 50.3). For comparison, the Qwen3-Embedding-0.6B + Faiss HNSW baseline achieves an average latency of 1.29 ms and nDCG@10 = 50.4. Seismic’s ANN inverted index with pruning therefore allows MILCO to (i) match this latency at $p = 10$ while achieving substantially higher effectiveness (nDCG@10 = 56.3), and (ii) further reduce latency to ≈0.44 ms at $p = 60$, while remaining at least as effective overall.
>
> **Table 2. Retrieval Latency of MILCO with Seismic index and Qwen3 Embedding 0.6 B Faiss Embedding.**
> | Model                  | Index / Method           | Avg Latency (ms) | P95 Latency (ms) | QPS         | nDCG@10   | Index Size |
> |------------------------|---------------------------|-------------------|-------------------|-------------|-----------|-----------|
> | Qwen3-Embedding-0.6B   | HNSW_M32_ef64             | 1.29             | 1.47             | 777         | 50.4    | 134G  |
> | MILCO (p=0)         | Seismic_qcut10_hf0.9      | 1.86               | 4.32              | 538         | 56.4         | 61G |
> | MILCO (p=10)           | Seismic_qcut10_hf0.9      | 1.29          | 2.92          | 774   | 56.3  | 40G |
> | MILCO (p=30)           | Seismic_qcut10_hf0.9      | 0.61         | 1.26          | 1647 | 54.4  | 25G |
> | MILCO (p=50)           | Seismic_qcut10_hf0.9      | 0.65          | 1.33          | 1545 | 53.3  | 16G |
> | MILCO (p=60)           | Seismic_qcut10_hf0.9      | 0.44          | 0.82          | 2265 | 50.3  | 12G |
>
> [1] Bruch et al., SIGIR 2024. Efficient Inverted Indexes for Approximate Retrieval over Learned Sparse Representations
>
> [2] Matthijs et al., arXiv 2024. The Faiss library

---

> ### Author Response · Authors · 2025-11-21
> **Response to Reviewer 4uhs (5/6)**
>
> > How does MILCO perform on truly low-resource languages not well represented in parallel corpus? Can the authors report per-language numbers for the lowest-performing languages?
>
> To address this question and measure the effect of parallel corpus used in sparse alignment pretraining, we report per-language ΔnDCG@10 between MILCO (alignment + distillation) and our best-performing dense baseline (distillation only) trained with the same data and compute (Table 1). The results are shown in Tables 3 and 4.
>
> Table 4 lists nDCG@10 and ΔnDCG@10 for all 18 MIRACL languages, including the lowest-performing ones, together with their proportion in the parallel corpus. Aggregating by coverage (Table 3), we observe an average gain of +1.59 ΔnDCG@10 for well-represented languages (P ≥ 1%) and a smaller but still positive average gain of +0.77  for under-represented languages (P < 1%), including languages with 0 parallel samples (sw, te, yo, bn). The main exception is fa (−1.09), which we plan to analyze further.
>
> Overall, these results suggest that MILCO provides consistent gains for well-represented languages and still yields smaller but positive improvements for languages that are weakly covered or entirely absent from the alignment corpus. This behavior highlights MILCO’s multilingual alignment pretraining as an effective mechanism for knowledge transfer and sharing across languages.
>
> **Table 3. The average Delta nDCG@10 between well-represented languages and under-represented languages in alignment data.**
>
> | Percentage Category:  |  P ≥ 1 | P < 1 |
> |---------------------|-------|-------|
> | Avg. Delta nDCG@10  | 1.59  | 0.77  |
>
>
> **Table 4. MILCO effectiveness vs. alignment data distribution across languages (MIRACL).**
> | Lang | nDCG@10 | Delta nDCG10 | Samples   | Percentage |
> |------|---------|--------------|-----------|------------|
> | ar   | 80.4    | 0.919        | 19002229  | 5.5        |
> | bn   | 82.6    | 2.387        | 0         | 0          |
> | en   | 60.4    | 3.585        | -         | -          |
> | es   | 60.9    | 0.487        | 28470451  | 8.23       |
> | fa   | 62.3    | -1.09        | 646913    | 0.19       |
> | fi   | 81.2    | 2.676        | 4958793   | 1.43       |
> | fr   | 61.7    | -0.922       | 22574836  | 6.53       |
> | hi   | 64.4    | 2.158        | 9626940   | 2.78       |
> | id   | 60.9    | 2.498        | 17591514  | 5.09       |
> | ja   | 77.2    | 2.524        | 11542221  | 3.34       |
> | ko   | 72.1    | 1.548        | 3648265   | 1.05       |
> | ru   | 74.6    | 2.458        | 16592711  | 4.8        |
> | sw   | 80.3    | 0.734        | 0         | 0          |
> | te   | 87.9    | 0.87         | 0         | 0          |
> | th   | 84.2    | 1.295        | 1668858   | 0.48       |
> | zh   | 65.5    | 1.328        | 9006690   | 2.6        |
> | de   | 61.4    | 1.823        | 24431450  | 7.07       |
> | yo   | 83.6    | 0.41         | 0         | 0          |

---

> ### Author Response · Authors · 2025-11-21
> **Response to Reviewer 4uhs (6/6)**
>
> > Can you provide more ablations on LexEcho fusion (e.g., varying weight between English and source views)
> To address this question, we add an ablation that interpolates between the pivot (English) and source views in LexEcho. Specifically, we multiply the English view by α and the source view by (1−α), with α ∈ {0.0, 0.2, …, 1.0}, and report MIRACL results in Table 5.
>
> We find that MILCO performs very poorly with only the source view (α = 0.0; average nDCG@10 = 5.61), indicating that the English pivot is essential. With only the English view (α = 1.0), MILCO is already strong (average nDCG@10 = 70.02) but not optimal. The best result is obtained with a roughly balanced fusion (α = 0.5), reaching an average nDCG@10 of 72.66 (around 3–4% relative improvement over α = 1.0). Performance for α between 0.5 and 0.6 is very similar, suggesting that LexEcho is not overly sensitive to the exact weighting as long as both views contribute.
>
> **Table 5. MILCO performance under different pivot–source weighting schemes on the MIRACL (hard negatives) benchmark.**
>
> | Alpha | ar    | bn    | en    | es    | fa    | fi    | fr    | hi    | id    | ja    | ko    | ru    | sw    | te    | th    | zh    | de    | yo    | Avg   |
> |-------|-------|-------|-------|-------|-------|-------|-------|-------|-------|-------|-------|-------|-------|-------|-------|-------|-------|-------|-------|
> | 0.00  | 5.21  | 9.61  | 3.56  | 0.09  | 1.74  | 8.18  | 0.14  | 1.56  | 4.71  | 3.33  | 11.14 | 2.19  | 2.75  | 31.35 | 14.90 | 0.00  | 0.20  | 0.27  | 5.61  |
> | 0.20  | 11.01 | 17.51 | 6.16  | 0.91  | 4.23  | 17.80 | 1.51  | 3.95  | 9.55  | 7.16  | 19.09 | 5.35  | 6.95  | 43.42 | 22.47 | 0.11  | 1.84  | 4.49  | 10.19 |
> | 0.40  | 79.38 | 79.30 | 56.55 | 55.89 | 57.29 | 79.72 | 60.70 | 57.46 | 58.46 | 75.36 | 71.99 | 73.10 | 77.52 | 86.77 | 82.44 | 56.95 | 59.67 | 63.23 | 68.43 |
> | 0.50  | 80.79 | 82.00 | 62.08 | 62.06 | 63.08 | 80.64 | 62.73 | 65.06 | 60.91 | 77.25 | 72.00 | 74.96 | 80.18 | 87.82 | 84.46 | 66.80 | 62.83 | 82.14 | 72.66 |
> | 0.60  | 80.08 | 81.43 | 62.03 | 62.00 | 62.34 | 79.91 | 63.14 | 64.94 | 60.14 | 76.64 | 71.13 | 74.30 | 79.61 | 87.11 | 83.81 | 65.54 | 61.73 | 81.72 | 72.09 |
> | 0.80  | 78.19 | 80.22 | 60.18 | 61.27 | 60.36 | 78.81 | 62.16 | 64.00 | 58.39 | 74.66 | 69.12 | 72.95 | 78.49 | 84.05 | 81.47 | 63.63 | 60.58 | 80.86 | 70.52 |
> | 1.00  | 77.77 | 79.35 | 59.53 | 60.98 | 59.73 | 78.43 | 62.19 | 63.31 | 57.97 | 73.80 | 68.52 | 72.74 | 78.04 | 82.90 | 80.72 | 63.33 | 60.55 | 80.46 | 70.02 |

---

> > ### Comment · Reviewer_4uhs · 2025-11-25
> >
> > Thanks for such detailed author responses. All of my concerns have been solved. It's refreshing to project other languages' tokens to English tokens as an effective method with strong multilingual LSR performances.
> >
> > I decide to raise my score to 6.
> >
> > Nit: MILCO is good with the existing English-dominant training sets. I also expect that the high-quality multilingual training data could grow larger. Maybe the size and quality of existing open-sourced multilingual training data are the fundamental keys.

---

> > > ### Author Response · Authors · 2025-11-27
> > >
> > > Dear reviewer 4uhs,
> > >
> > > Thank you for the follow-up and for raising your score. We believe that MILCO’s performance could be further improved with larger and higher-quality multilingual training data, and we see this as an important direction for future work.
> > >
> > > Kind regards,
> > > The authors

---

### Official Review · Reviewer_LS1r · 2025-11-01

**Soundness:** 3
**Presentation:** 3
**Contribution:** 3
**Rating:** 6
**Confidence:** 2

**Summary:**

This paper introduces MILCO, a LSR architecture for multilingual and cross-lingual retrieval. MILCO maps queries and documents into a shared English lexical space through the integration of a multilingual connector and a LexEcho head, enhancing robustness to uncommon entities. The paper also proposes a SAP strategy, which addresses the entity loss issue and improves retrieval effectiveness. Experimental results show that MILCO outperforms existing methods on multilingual and cross-lingual benchmarks, demonstrating high efficiency and accuracy.

**Strengths:**

1. The paper identifies that during cross-lingual projection, entities from non-Latin languages (e.g., Chinese or Arabic proper nouns) are often lost. To address this, it introduces the LexEcho head, which generates dual sparse representations — one from the English view and one from the source-language view — significantly improving robustness to rare entities and enhancing cross-lingual consistency.

2. The paper introduces Sparse Alignment Pretraining (SAP), a novel pretraining strategy specifically designed for multilingual LSR. Unlike prior methods that align multilingual inputs within dense semantic spaces, SAP directly aligns multilingual inputs with English lexical targets at the vocabulary level.

3. The paper is well-written.

**Weaknesses:**

Although MILCO improves training efficiency by mapping multilingual inputs into a unified English lexical space via the “Multilingual Connector,” this approach may introduce an English-centric bias. For languages with syntactic structures or lexical mappings that differ substantially from English, such monolingual alignment could lead to semantic distortion or representational misalignment, raising concerns about its generalization and fairness across diverse linguistic systems.

**Questions:**

N/A

---

> ### Author Response · Authors · 2025-11-21
> **Response to  Reviewer LS1r (1/1)**
>
> > Although MILCO improves training efficiency by mapping multilingual inputs into a unified English lexical space via the “Multilingual Connector,” this approach may introduce an English-centric bias. For languages with syntactic structures or lexical mappings that differ substantially from English, such monolingual alignment could lead to semantic distortion or representational misalignment, raising concerns about its generalization and fairness across diverse linguistic systems.
>
> Following the LSR paradigm of producing representations grounded in a lexical vocabulary, we aim to produce a universal LSR model for both monolingual and cross-lingual retrieval across many languages, hence there needs to be a pivot language.  MILCO is, in principle, pivot-language agnostic: the Multilingual Connector can project into any lexical space given sufficient alignment data. We choose English purely for practical reasons, as it is the only language for which large-scale parallel data between a pivot language and other languages is widely available. We agree that any pivot (including English) may introduce bias or semantic distortion, and the lost entity phenomenon we analyze is one concrete example of such misalignment. To mitigate this, MILCO uses the LexEcho head, which combines both the projected English view and the original-language view, allowing the model to recover information that might be distorted by the pivot. Empirically, this dual-view design improves robustness and overall retrieval effectiveness; we will clarify this motivation and its implications for bias and generalization in the revised version.
>
> We also wish to highlight that such bias detection and fix as in LexEcho is possible with MILCO thanks to its representation transparency allowing us to interpret, inspect and detect issues in representations (e.g., the un-popular entity loss issue was identified while we were manually looking at the projection output). Language biases and misalignment do exist in dense retrieval systems [1, 2], however it is much more difficult to detect them due to the black-box nature of dense representations.  In addition to the transparency advantage, MILCO also exhibits strong retrieval effectiveness compared to dense baselines, even without the LexEcho head,  and offers training-free post-hoc pruning that significantly speeds up retrieval latency and index size.
>
> [1] Jinrui et al., EMNLP MRL 2024.  Language Bias in Multilingual Information Retrieval: The Nature of the Beast and Mitigation Methods
>
> [2] Eugene et al., SIGIR 2024. Language Fairness in Multilingual Information Retrieval

---

> > ### Author Response · Authors · 2025-11-27
> >
> > Dear reviewer LS1r,
> >
> > We have provided detailed response to your concern about biases and mis-alignment in cross-lingual projection. We would greatly appreciate it if you could let us know whether these answers resolve your concern.
> >
> > Best regards,
> >
> > The authors

---

### Official Review · Reviewer_ej1c · 2025-11-02

**Soundness:** 3
**Presentation:** 3
**Contribution:** 2
**Rating:** 4
**Confidence:** 4

**Summary:**

This paper proposes a novel method for a learned sparse retriever (LSR) model targeting multilingual and cross-lingual retrieval tasks, an area where LSRs typically struggle. Dense embedding retrievers usually perform better on such tasks due to cross-lingual lexical and entity mismatches, which make sparse methods harder to optimize. The authors introduce a LexEcho head that maps sparse representations into an English pivot space while preserving the original language tokens with predicted weights for indexing. This multilingual fusion representation is shown to be crucial through ablation studies. The final model is trained using language alignment pre-training and contrastive learning with distillation. Experimental results on both multilingual and cross-lingual benchmarks demonstrate the effectiveness of the proposed LSR method.

**Strengths:**

1. The proposed method is creative and effective.
2. The ablation studies on the modeling components are insightful and clearly demonstrate the importance of each part of the method.

**Weaknesses:**

1. The paper evaluates efficiency only through token pruning. However, it does not report the actual retrieval latency when using the sparse index, nor compare it to the latency of dense embedding models. Fewer tokens or sparsity does not necessarily guarantee faster retrieval, since inverted indices may produce very long posting lists in multilingual settings.
2. A baseline dense embedding model trained with the same language alignment pre-training and contrastive learning with distillation (from the same cross-encoder and data) would strengthen the comparison of sparse vs dense. The paper presents an effective LSR model for multilingual retrieval. However, the weaknesses highlight missing experimental comparisons that make it difficult to determine whether sparse or dense models are ultimately more suitable for this task.

**Questions:**

- How does the proposed model compare to the dense retrieval model under the same training setup?
- How does the sparse retriever model with token pruning compare to the dense model running on GPU on latency?

---

> ### Author Response · Authors · 2025-11-21
> **Response to Reviewer ej1c (1/2)**
>
> > How does the proposed model compare to the dense retrieval model under the same training setup?
>
> Table 1 compares MILCO to dense retrieval baselines trained under exactly the same conditions on MIRACL evaluation benchmark: using the same backbone, same data, and the same two-stage pipeline of alignment followed by cross-encoder distillation.
>
> For the aligned dense baseline, we use **thenlper/gte-base** as the English teacher because it is similar in size and BEIR (English) performance to our **SPLADE-v3** sparse English teacher (GTE-base: 52.61 nDCG@10, SPLADE-v3: 51.69 nDCG@10). We align the dense English and multilingual representations with an MSE loss, following [1]; since dense vectors are mostly non-zero (not sparse), this is effectively equivalent to the SparseMSE loss used for MILCO. The distillation step is identical to MILCO’s, but without sparse regularization. This dense model achieves an average nDCG@10 of **67.9** on MIRACL (hard negatives), compared to **72.7** for MILCO under the same alignment + KL distillation setup. WIth KL distillation, we use the reranker (**bge-reranker-v2.5-gemma2-lightweight**)  for both sparse and dense models.
>
> We also train a dense baseline **without** the alignment step, using only KL distillation from the same cross-encoder reranker. This model performs better than the aligned dense baseline (average nDCG@10 = **70.9**) and is comparable to MILCO trained with alignment + InfoNCE and in-batch negatives, but still underperforms our best MILCO variant with alignment + KL distillation. Both dense baselines and MILCO are initialized from **BAAI/bge-m3-unsupervised** [2], which is pretrained on large-scale retrieval data, including parallel sentences used in our sparse alignment step. We found that alignment pretraining is critical to prevent semantic collapse for multilingual sparse retrieval, but is not necessary for dense model training.
>
> Beyond effectiveness, we want to also highlight that MILCO’s sparse lexical representations offer practical advantages. They are transparent, enabling manual inspection and targeted fixes (e.g., our LexEcho head correcting entity-related errors), and they support post-hoc pruning without requiring a Matryoshka-style nested loss, allowing dynamic control of index size and latency. Theoretically, Orion et al. [3] show that sparse lexical embeddings have higher representational capacity than dense embeddings; we validated that MILCO’s lexical embeddings outperforms strong dense baselines on the LIMIT test proposed in [3].
>
> **Table 1. Comparison between MILCO and dense baselines under the same training setup (Alignment + Distillation), evaluated on MIRACL benchmark (hard negatives).**
> | Method                                | Size | ar   | bn   | en   | es   | fa   | fi   | fr   | hi   | id   | ja   | ko   | ru   | sw   | te   | th   | zh   | de   | yo   | avg |
> |---------------------------------------|------|------|------|------|------|------|------|------|------|------|------|------|------|------|------|------|------|------|------|------|
> | Dense (Alignment + KL Distillation)   | 561M | 77.0 | 76.6 | 55.3 | 57.5 | 60.2 | 77.1 | 59.0 | 60.5 | 55.5 | 70.5 | 70.9 | 67.5 | 74.4 | 86.1 | 80.6 | 62.5 | 57.6 | 72.8 | 67.9 |
> | Dense (KL Distillation)               | 561M | 79.5 | 80.2 | 56.8 | 60.4 | 63.4 | 78.5 | 62.6 | 62.2 | 58.4 | 74.7 | 70.5 | 72.1 | 79.6 | 87.0 | 82.9 | 64.2 | 59.5 | 83.2 | 70.9 |
> | MILCO (Alignment + InfoNCE)           | 561M | 79.7 | 81.1 | 57.5 | 57.5 | 60.8 | 80.4 | 58.0 | 63.2 | 58.7 | 75.4 | 71.3 | 72.8 | 77.5 | 87.1 | 83.0 | 61.0 | 59.5 | 81.8 | 70.3 |
> | MILCO (Alignment + KL Distillation)   | 561M | 80.8 | 82.0 | 62.1 | 62.1 | 63.1 | 80.6 | 62.7 | 65.1 | 60.9 | 77.3 | 72.0 | 75.0 | 80.2 | 87.8 | 84.5 | 66.8 | 62.8 | 82.1 | 72.7 |
>
> [1] Nils et al., EMNLP 2020. Making Monolingual Sentence Embeddings Multilingual using Knowledge Distillation.
>
> [2] Jianlv Chen et al., ACL 2024. M3-Embedding: Multi-Linguality, Multi-Functionality, Multi-Granularity Text Embeddings Through Self-Knowledge Distillation
>
> [3] Orient et al. On the Theoretical Limitations of Embedding-Based Retrieval

---

> ### Author Response · Authors · 2025-11-21
> **Response to Reviewer ej1c (2/2)**
>
> > How does the sparse retriever model with token pruning compare to the dense model running on GPU on latency?
>
> We compare the query latency between MILCO (560M) and Qwen3-Embedding-0.6B. We decompose the total query latency into (1) query encoding latency on GPU and (2) retrieval latency from an index on CPU. Since MILCO and Qwen3 have similar model sizes, the GPU encoding time to obtain representations from raw text is comparable for both models (~1.86 ms/query on one AMD MI250x GPU). Therefore, the difference in query latency mainly comes from the retrieval stage.
>
> For indexing and retrieval, we use Seismic [4] for MILCO and Faiss [5] for Qwen3-Embedding-0.6B on the MIRACL English collection with ~32 million passages. To ensure a fair comparison, we follow prior work [4, 6, 7] and run both systems on a single CPU core under identical hardware specifications (AMD EPYC 7763 64-Core) and memory, and report latency and effectiveness for different index types in Table 2.
>
> For Qwen3-Embedding-0.6B, we experiment with three index types (Flat, IVFFlat, and HNSW). Among them, the Flat index, which performs exhaustive lossless search with no index compression, obtains the highest retrieval effectiveness (nDCG@10 = 51.6) but incurs extremely high retrieval latency (4470.52 ms/query). Dense approximate nearest neighbor (ANN) methods such as IVFFlat and HNSW offer significant speed-ups at the cost of some accuracy loss. We tried different configurations and found that HNSW (M = 32, ef = 64) achieves the best trade-off, with an average latency of 1.28 ms/query and nDCG@10 of 50.4.
>
> For MILCO, we measure latency at different pruning levels of the English view (p ∈ {0, 10, 30, 50, 60}). With no pruning (p = 0, average posting list length: 4636.09), MILCO is slower than Qwen3-Embedding-0.6B with an HNSW index, but achieves a much higher nDCG@10 of 56.0. When pruning the lowest-weight dimensions whose cumulative weights account for 10% of the total, MILCO with Seismic achieves a 30% speed-up, resulting in comparable latency (1.29 ms/query) to Faiss HNSW with almost no effectiveness degradation. With 30% pruning, MILCO is twice as fast while still more effective than Qwen3-Embedding-0.6B with HNSW. With an aggressive 60% pruning, MILCO remains as effective as Qwen3-Embedding-0.6B (1024 dimensions) while being three times faster.
>
> These results confirm the advantage of MILCO: post-hoc pruning can significantly reduce retrieval latency, and sparse ANN search (e.g., Seismic) can be competitive with dense ANN search (e.g., Faiss). Pruning the sparse model not only reduces latency but also dramatically decreases the required **index size**. For instance, aggressive 60% pruning ($p=60$) shrinks the MILCO index to just **12GB**, which is more than $11\times$ smaller than the Qwen3-HNSW dense index (**134GB**), making it advantageous for memory-constrained deployment environments.
>
>
> **Table 2. Retrieval Latency of MILCO with Seismic index and Qwen3 Embedding 0.6 B Faiss Embedding.**
> | Model                  | Index / Method           | Avg Latency (ms) | P95 Latency (ms) | QPS         | nDCG@10   | Index Size |
> |------------------------|---------------------------|-------------------|-------------------|-------------|-----------| -----------|
> | Qwen3-Embedding-0.6B   | Flat                      | 4470.52         | 4518.32          | 0           | 51.6     | * |
> | Qwen3-Embedding-0.6B   | IVF256_Flat_nprobe16      | 287.96           | 366.37           | 3           | 50.3    | * |
> | Qwen3-Embedding-0.6B   | HNSW_M32_ef64             | 1.29             | 1.47             | 777         | 50.4    | 134G  |
> | MILCO (p=0)         | Seismic_qcut10_hf0.9      | 1.86               | 4.32              | 538         | 56.4         | 61G |
> | MILCO (p=10)           | Seismic_qcut10_hf0.9      | 1.29          | 2.92          | 774   | 56.3  | 40G |
> | MILCO (p=30)           | Seismic_qcut10_hf0.9      | 0.61         | 1.26          | 1647 | 54.4  | 25G |
> | MILCO (p=50)           | Seismic_qcut10_hf0.9      | 0.65          | 1.33          | 1545 | 53.3  | 16G |
> | MILCO (p=60)           | Seismic_qcut10_hf0.9      | 0.44          | 0.82          | 2265 | 50.3  | 12G |
>
> [4] Bruch et al., SIGIR 2024. Efficient Inverted Indexes for Approximate Retrieval over Learned Sparse Representations
>
> [5] Matthijs et al., arXiv 2024. The Faiss library
>
> [6] Carlos et al., SIGIR 2022. An Efficiency Study for SPLADE Models
>
> [7] Yingrui et al., ECIR 2025. LSTM-Based Selective Dense Text Retrieval Guided by Sparse Lexical Retrieval

---

> > ### Author Response · Authors · 2025-11-27
> > **Following-up Response**
> >
> > Dear reviewer ej1c,
> >
> > We have provided detailed responses to your questions about (1) how MILCO compares to dense retrieval under the same training setup and (2) how MILCO with token pruning compares to a dense model in terms of latency, and we would greatly appreciate it if you could let us know whether these answers resolve your concerns.
> >
> > Best regards,
> >
> > The authors

---

> > > ### Comment · Reviewer_ej1c · 2025-11-27
> > >
> > > Thanks for the response.
> > > For Q1, thanks for the clarification.
> > > For Q2, what I originally meant is how does the dense retriever search on faiss-gpu compares. (It would be good to see but I don't think it is major issue)
> > > Thanks for the detailed response. I do not have major concern with the paper. I have increased the score.

---

> > > > ### Author Response · Authors · 2025-12-02
> > > >
> > > > > Thanks for the response. For Q1, thanks for the clarification. For Q2, what I originally meant is how does the dense retriever search on faiss-gpu compares. (It would be good to see but I don't think it is major issue) Thanks for the detailed response. I do not have major concern with the paper. I have increased the score.
> > > >
> > > > Thank you for the clarification on Q2 and for increasing your score.
> > > >
> > > > We have now added an explicit comparison with dense retrieval using Faiss on an A100 40GB GPU (Table 3). Note that **faiss-gpu** [8] only supports exhaustive search with **IndexFlat**, **IndexIVFFlat** or simple partition-based search with  **IndexIVFScalarQuantizer** (SQ), and **IndexIVFPQ** (PQ) indexes.  Advanced graph-based algorithms, such as **HNSW**, are not available on GPUs.
> > > >
> > > > In our ~32M-document setting, Flat and IVF-Flat indexes for Qwen3-Embedding-0.6B do not fit on a single A100 40GB GPU (they would require ~140GB of VRAM), and Faiss-GPU does not support disk-backed offloading. We therefore evaluated compressed GPU indexes (IVF-SQ and IVF-PQ). With moderate compression (IVF8192_SQ8, nprobe=16), GPU search achieves 146 ms/query, which is roughly two orders of magnitude slower than MILCO+Seismic on CPUs (0.44–1.86 ms/query), and still yields lower nDCG@10. Using more aggressive  PQ compression (e.g., IVF8192_PQ64) reduces GPU latency to ~1.4 ms/query, but nDCG@10 drops to 40.5, clearly below MILCO’s 50–56 range. We also tried searching on these quantized indexes with higher *nprobe*, but the effectiveness gains were negligible and still significantly below MILCO.
> > > >
> > > > Overall, in our 32M-document setup and with the Faiss-GPU index types we evaluated, Qwen3-Embedding-0.6B on GPU either cannot fit in memory or must be heavily quantized, while MILCO+Seismic on commodity CPUs offers a strictly better effectiveness–efficiency trade-off.
> > > >
> > > > **Table 3. Retrieval Latency of MILCO with Seismic index (CPU) and Qwen3 Embedding 0.6 B Faiss (CPU, GPU).**
> > > >
> > > > | Model                   | Index / Method                    | Avg Latency (ms) | P95 Latency (ms) | QPS  | nDCG@10 | Index Size |
> > > > |-------------------------|-----------------------------------|------------------|------------------|------|---------|------------|
> > > > | Qwen3-Embedding-0.6B    | HNSW_M32_ef64   (**CPU**)                  | 1.29             | 1.47             | 777  | 50.4    | 134G       |
> > > > | Qwen3-Embedding-0.6B    | IVF8192_SQ8_nprobe16 (**GPU**)        | 146.20           | 395.00           | 6    | 48.1    | 32G        |
> > > > | Qwen3-Embedding-0.6B    | IVF8192_SQ8_nprobe64 (**GPU**)        | 249.50           | 425.70           | 4    | 50.4    | 32G        |
> > > > | Qwen3-Embedding-0.6B    | IVF8192_PQ32_nprobe16 (**GPU**)       | 1.02             | 2.10             | 973  | 34.0    | 1.3G       |
> > > > | Qwen3-Embedding-0.6B    | IVF8192_PQ64_nprobe16 (**GPU**)       | 1.40             | 2.70             | 697  | 40.5    | 2.3G       |
> > > > | MILCO (p=0)             | Seismic_qcut10_hf0.9   (**CPU**)            | 1.86             | 4.32             | 538  | 56.4    | 61G        |
> > > > | MILCO (p=10)            | Seismic_qcut10_hf0.9   (**CPU**)              | 1.29             | 2.92             | 774  | 56.3    | 40G        |
> > > > | MILCO (p=30)            | Seismic_qcut10_hf0.9   (**CPU**)              | 0.61             | 1.26             | 1647 | 54.4    | 25G        |
> > > > | MILCO (p=50)            | Seismic_qcut10_hf0.9   (**CPU**)              | 0.65             | 1.33             | 1545 | 53.3    | 16G        |
> > > > | MILCO (p=60)            | Seismic_qcut10_hf0.9   (**CPU**)              | 0.44             | 0.82             | 2265 | 50.3    | 12G        |
> > > >
> > > >
> > > > [8] https://github.com/facebookresearch/faiss/wiki/Faiss-on-the-GPU

---

### Author Response · Authors · 2025-11-21
**General response to reviewers**

We thank all reviewers for their time and thoughtful evaluations of our work. We are happy to see that reviews consistently highlighted that our method is creative, effective, and clearly presented, with novel and well-motivated technical contributions supported by our empirical evaluation. In the following part of this rebuttal, we reply and address the raised concerns in separate responses to each individual reviewer.

---

### Author Response · Authors · 2025-12-03
**Summary of Rebuttal Discussion**

Dear Area Chair,

We are encouraged that the reviewers consistently highlighted that our method is creative, effective, and clearly presented, with novel and well-motivated technical contributions supported by our strong empirical evaluation. We write to briefly summarize our engagement during the rebuttal period.

We are pleased to report that we have successfully addressed the concerns raised by Reviewer ej1c and Reviewer 4uhs, resulting in positive feedback and score increases (from 4 to 6) from both. We would also like to explicitly note that these discussions and the subsequent score increases were finalized **prior to or concurrent with** the widespread news regarding the reviewer identity leakage incident. Specifically, the issue was formally reported by the ICLR 2026 Workflow Chair at **10:09 AM EST on November 27**, whereas Reviewer 4uhs increased their score well before this (**at 4:36 AM EST, Nov 25**) and Reviewer ej1c increased their score shortly thereafter (**at 10:51 AM EST, Nov 27**).

**Key rebuttal highlights:**

1. **Latency and Efficiency Analysis (Reviewers ej1c & 4uhs):**
We provided a detailed latency comparison between MILCO (using the Seismic index) and Qwen3-Embedding (using Faiss HNSW). The results show that with 60% token pruning, MILCO is 3x faster and requires 10x less index storage (12GB vs. 134GB) than Qwen3-Embedding, while maintaining comparable effectiveness.

2. **Comparison with Dense Baselines (Reviewers ej1c & 4uhs)**
We conducted an apples-to-apples comparison against dense retrieval models under the same training setup (using the same backbone, data, and alignment/distillation setup). MILCO (72.7 nDCG@10) outperforms the equivalent dense baseline (67.9 nDCG@10) on MIRACL, demonstrating that our performance gains come from the method itself, not just training data.

3. **Low-Resource Languages & Ablation Studies (Reviewer 4uhs)**
We analyzed MILCO’s performance on low-resource languages, showing positive gains even for languages with zero parallel training samples.  We also provided an ablation on the LexEcho fusion weighting, confirming that a balanced fusion of English and source views yields the best performance and that LexEcho is not overly sensitive to the weighting parameter.

We have also provided a revised manuscript in which these rebuttal points have been incorporated; the corresponding changes are highlighted in blue.

---

### Meta-Review · Area_Chair_Nncm · 2026-01-06

**Summary:**

This paper proposes MILCO, a multilingual sparse retrieval model designed to address the issues of non-Latin entity loss and semantic collapse common in cross-lingual projection. The method innovatively introduces Sparse Alignment Pretraining (SAP) and the LexEcho module. The former eschews traditional dense space alignment by directly mapping multilingual inputs into a unified English lexical space, while the latter enhances the capability to capture rare entities by constructing dual sparse views of the English and source languages. Overall, the paper aims to significantly improve the robustness and accuracy of multilingual retrieval via this English-hub lexical alignment strategy, while maintaining the efficiency of sparse indexing.
**Strengths:**
1.	The proposed SAP and LexEcho effectively resolve the semantic collapse issue in cross-lingual logit mapping and significantly improve robustness regarding rare entities.
2.	The model demonstrates superior performance across multiple benchmarks, and the ablation studies clearly validate the individual contributions of each component.
3.	The paper is well-structured and effectively articulates the practical challenges associated with multilingual sparse retrieval.


**Weaknesses:**
1.	The paper lacks a fair comparison with dense baselines under identical training settings, making it difficult to definitively ascertain the actual advantages of the sparse model over dense models in multilingual tasks. Furthermore, the paper fails to report actual query latency and index sizes.
2.	The study lacks rigor due to the non-disclosure of training computational costs, specific details regarding the teacher model, and potential dataset overlap.
3.	There is an absence of robustness analysis for low-resource scenarios. Additionally, ablation studies on key hyperparameters, such as the LexEcho fusion weights, are missing.
4.	Forcing the mapping of all multilingual tokens into the English lexical space lacks sufficient justification and may introduce an English-centric bias.

**Reviewer Concerns:**

1.	The three major concerns raised by Reviewer ej1c，specifically, the lack of a fair comparison with dense models under identical training settings, the absence of actual retrieval latency reporting, and the latency comparison against dense models on GPUs，have been fully addressed by the authors' experiments. Accordingly, the reviewer explicitly confirmed in their follow-up response: 'I do not have major concern with the paper.'
2.	The concern regarding 'English-centric bias' raised by Reviewer LS1r has been addressed by the authors through literature citations and theoretical explanation, although the response lacks specific experimental support.
3.	The major concerns raised by Reviewer 4uhs，specifically regarding the motivation and rationale for mapping to the English lexical space, the necessity of the method in non-cross-lingual tasks, the rigor of the experimental comparisons, and the ablation studies for LexEcho, have all been successfully resolved through the authors' detailed experiments and responses.

**Reviewer Scores:**

I anticipate that Reviewers ej1c and 4uhs will likely upgrade their overall ratings from 4 to 6. This projection is based on the fact that the authors have comprehensively addressed these reviewers' primary concerns through detailed responses and substantial supplementary experimental evidence.

Conversely, I predict that Reviewer LS1r will maintain their original overall rating (6). The reservations raised by this reviewer regarding the novelty of the work and specific methodological drawbacks were not sufficiently mitigated during the rebuttal. The authors' response to these points relied heavily on theoretical justification rather than providing the persuasive, rigorous experimental validation necessary to fully resolve the concerns.

---

### Decision · Program_Chairs · 2026-01-26

Accept (Poster)